# Non-flammable solvent-free liquid polymer electrolyte for lithium metal batteries

Guo-Rui Zhu[1], Qin Zhang[1], Qing-Song Liu[1], Qi-Yao Bai[1], Yi-Zhou Quan[1], You Gao[1], Gang Wu [1] ✉ & Yu-Zhong Wang [1] ✉

As a replacement for highly flammable and volatile organic liquid electrolyte, solid polymer electrolyte shows attractive practical prospect in high-energy lithium metal batteries. However, unsatisfied interface performance and ionic conductivities are two critical challenges. A common strategy involves introducing organic solvents or plasticizers, but this violates the original intention of security design. Here, an electrolyte concept called liquid polymer electrolyte without any small molecular solvents is proposed for safe and high-performance batteries, based on the design of a room-temperature liquid-state brush-like polymer as the sole solvent of lithium salts. This liquid polymer electrolyte is non-flammable and exhibits high ionic conductivity ($1.09 \times 10^{-4}$ S cm$^{-1}$ at 25 °C), significant lithium dendrite suppression, and stable long-term cycling over a wide operating temperature range ( ≥ 1000 cycles at 60 °C and 90 °C). Moreover, the pouch cell can resist thermal abuse, vacuum environment, and mechanical abuse. This electrolyte and design strategy are expected to provide enlightening ideas for the development of safe and high-performance polymer electrolytes.

Lithium batteries (LBs) have revolutionized modern energy storage devices since their commercialization in 1991[1,2]. However, they have long been limited to use at around room temperature (RT) due to material limitations and safety concerns[3,4]. Currently, cutting-edge scientific research urgently calls for batteries with working temperatures covering a wide range[4,5] (i.e., below 0 °C and above 60 °C). However, working at high temperatures (HT, over 60 °C) will severely degrade the performance of the LBs and even cause thermal runaway due to the intrinsic thermolabilities of the electrodes, electrolytes, and separators[6]. Clearly, innovations in the LB materials are the ultimate way to avoid this operating temperature bottleneck[4].

Organic liquid electrolytes (LEs) have been widely used for conventional LBs due to their high ionic conductivities and their excellent and stable interfacial wettabilities[7]. However, the ethers and carbonate solvents commonly used in LBs have low boiling points and high flammability[8,9]. When chronically exposed to high temperatures, LEs can induce the LBs to swell, rupture and leak and even cause fire and explosion[7,10]. In addition, for lithium metal anodes, anabatic lithium dendrite growth and uncontrollable side reactions occur at the interface between the lithium metal and the LE at high temperatures and further compromise safety[11–15]. As an alternative, safer solid-state electrolytes have received considerable attention[4,5]. Although solid ceramic electrolytes are highly thermotolerant and fireproof[16], some critical issues remain to be resolved, including the low oxidation stability, sensitivity to H$_2$O or O$_2$, and poor interface compatibility[17,18]. In comparison, solid polymer electrolytes (SPEs) overcome the above shortcomings and offer shape versatility, flexibility, lightweight, and low-cost processing[19,20]. Many polymers have served as the matrixes of SPEs, such as poly(ethylene oxide) (PEO)[21,22], poly(propylene carbonate) (PPC)[23,24], and poly(1,3-dioxolane) (PDOL)[25,26]. However, low RT ionic conductivities and poor interfacial contacts are still the key challenges to practical application of SPE-based LBs[19]. The transfer of lithium ions (Li$^+$) depends on chain movement and the coordination/dissociation of Li$^+$ with polar atoms in the SPEs[19]. Therefore, polymers

[1]The Collaborative Innovation Center for Eco-Friendly and Fire-Safety Polymeric Materials (MoE), National Engineering Laboratory of Eco-Friendly Polymeric Materials (Sichuan), State Key Laboratory of Polymer Materials Engineering, College of Chemistry, Sichuan University, Chengdu 610064, China. ✉e-mail: gangwu@scu.edu.cn; yzwang@scu.edu.cn

serving as the matrixes are required to have low glass transition temperatures and abundant ion-conducting groups. Moreover, a highly flexible and deformable interface between the electrolyte and electrode is also necessary since a significant volume change occurs at the anode during cycling. However, the poor deformabilities and wettabilities of SPEs can lead to interfacial voids and inhomogeneous deposition and induce Li dendrite growth[27]. As a compromise, quasi-solid or gel polymer electrolytes (QSPEs or GPEs) containing organic small-molecular plasticizers/solvents have been used to improve the interface contact[28] and RT ionic conductivity[23,29,30]. However, their susceptibilities to thermal runaway at high temperatures restrict potential application[31].

When the polymer is in a molten or viscous state, it can also flow like a small-molecular liquid. Inspired by this feature, we propose an innovative electrolyte concept called liquid polymer electrolytes (LPEs), and the LPEs are composed of lithium salts and a liquid-state polymer as the sole solvent. The LPEs are expected to combine or even surpass the advantages of LEs and SPEs (Fig. 1a). The key to realizing this concept is to design a functional polymer with a low glass transition temperature, weak intermolecular forces, and low chain entanglement. Comb- or brush-shaped polymers are very promising candidates because their dense, short side chains and steric hindrance effects should prevent entanglement of side chains and main chains[19,32,33].

Herein, we report a nonflammable LPE without any other small molecular solvent or plasticizer to achieve excellent cyclability and all-around safety for lithium metal batteries (LMBs); a room-temperature liquid-state brush-like polymer consisting of flame-retardant polyphosphazene as the backbone and methoxytriethoxy substituents as the side chains was designed, i.e., poly[bis-(methoxytriethoxy) phosphazene] (PPZ), and used as the sole solvent for a common commercialized lithium salt. This LPE exhibits multiple advantages compared with conventional LEs and SPEs (Fig. 1b), i.e., (1) abundant N and P elements in the PPZ backbone ensure nonflammability via both gas- and solid-phase flame retardant models; (2) PPZ and its derived $Li_3N$ and $Li_3PO_4$ endow the solid electrolyte interface (SEI) with excellent thermal stability, mechanical strength, ionic conductivity, and flexibility to effectively suppress lithium dendrite growth and electrolyte-lithium side reactions; (3) coordination of the polar N and O atoms in the PPZ with $Li^+$ facilitates the dissociation of LiTFSI to form carriers and transfer of $Li^+$ through the movements of the phosphazene backbone and ether side chains; and (4) the flowable viscous LPE fully wets and even infiltrates the electrodes to maintain excellent contact and integrity of the interfacial layer but does not leak due to high temperatures or mechanical abuse. As a result, Li//Li symmetrical cells with this LPE enable uniform Li plating/stripping for over 2200 h at 90 °C without lithium dendrite growth. Remarkably, stable long-term cycling

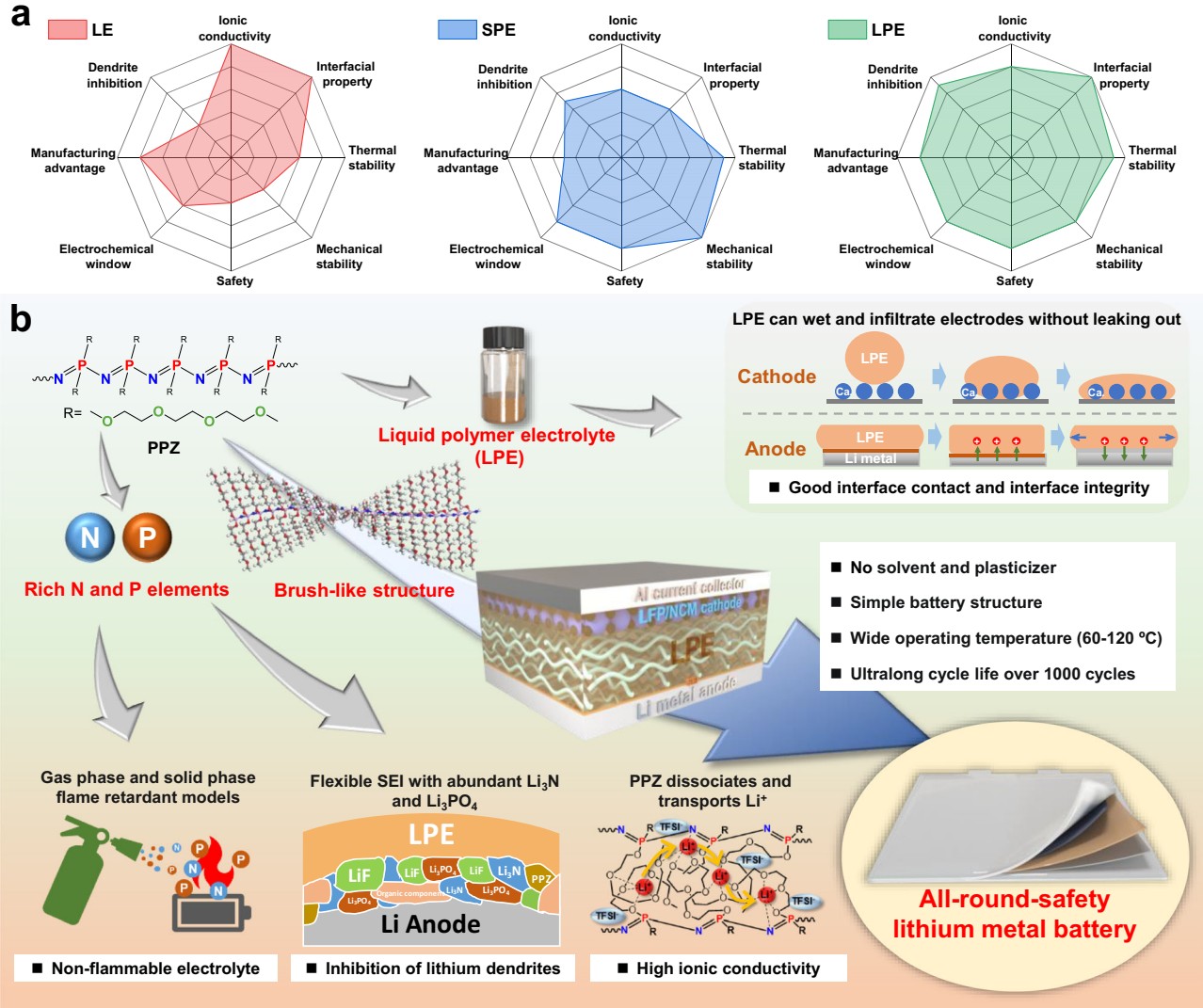

**Fig. 1 | Characteristics of liquid polymer electrolytes (LPEs) for all-around-safety lithium metal batteries. a** Radar plots of comprehensive comparison of LE, SPE, and LPE. **b** Schematic illustration of multifunctional LPEs.

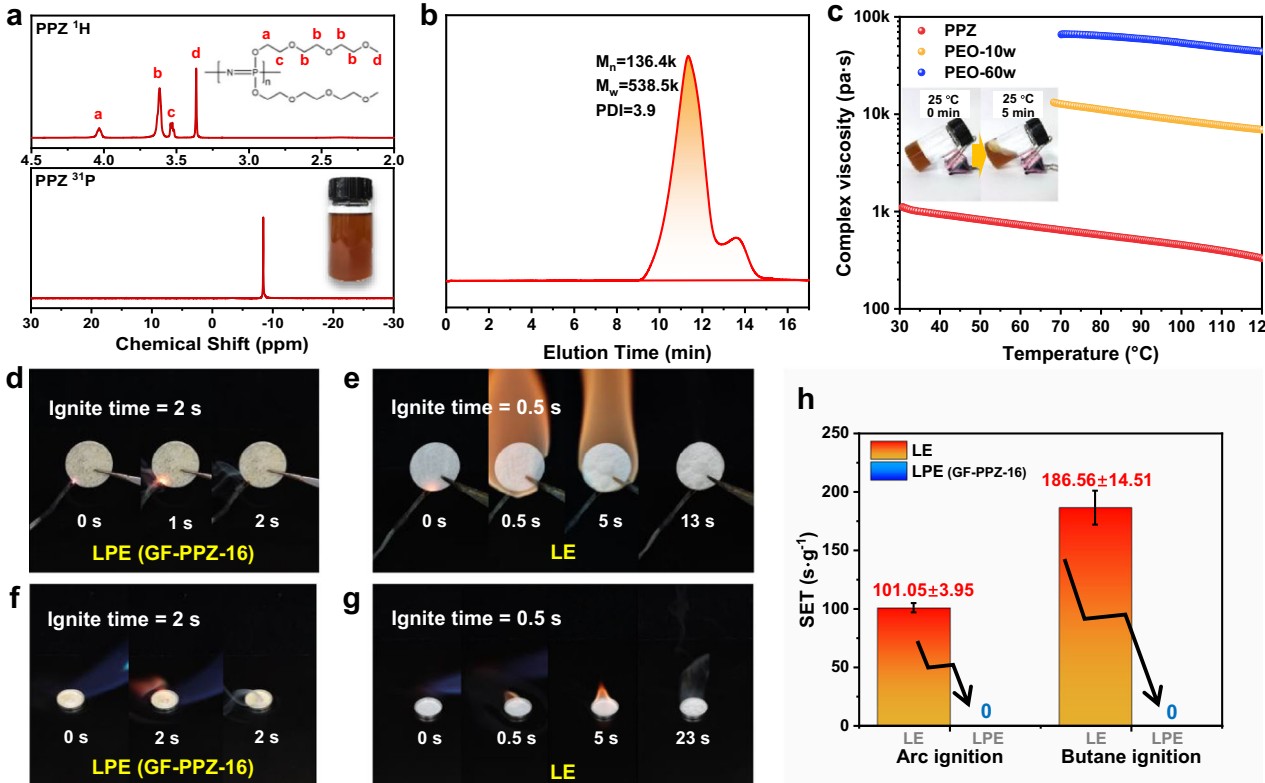

**Fig. 2 | Characterization of PPZ polymer. a** ¹H NMR and ³¹P NMR spectra (The inset is the digital photo of PPZ) and (**b**) GPC chromatogram of PPZ. **c** Rheological behavior of PPZ and PEO (The inset show that PPZ can flow at RT). Combustion test of LPE (GF-PPZ-16) and LE, **d, e** arc ignition and (**f, g**) butane ignition. **h** Self-extinguishing time (SET) histogram of LPE (GF-PPZ-16) and LE under two ignition modes (Bars denote S.D., GF only as support.). Source data are provided as a Source Data file.

of the LiFePO₄ and NCM811 cathodes can be achieved from 60 to 120 °C with a coulombic efficiency of approximately 100% and capacity fading of 0.038% per cycle over 1000 cycles. Moreover, the resistances to thermal shock, vacuum environments, and mechanical abuse also demonstrate the overall safety of the LPE-based LMBs.

## Results

### Preparation and characterization of liquid polymer electrolytes

To obtain LPEs, poly[bis-(methoxytriethoxy) phosphazene] (PPZ) was synthesized by melt polymerization of a phosphonitrilic chloride trimer (HCCP) followed by side-group substitution. As shown in Fig. 2a–b, the as-obtained PPZ with a weight of approximately 25 g is a highly viscous brown liquid at RT and presents a well-defined chemical structure with a large number-average molecular weight (Mₙ) of 136.4 kDa according to nuclear magnetic resonance (NMR) and gel permeation chromatography (GPC) tests. Rheological measurements showed that the PPZ has a low and relatively stable viscosity over the wide temperature range 30–120 °C (Fig. 2c) and even flows at RT (the inset of Fig. 2c). As a comparison, semicrystalline PEO is a solid power at RT and has very high viscosities above 70 °C.

The LPEs, which did not contain any small molecular solvent, plasticizer, or other additive, were prepared by dissolving a commercial lithium salt into the PPZ. The as-prepared LPEs containing different contents of lithium bis(trifluoromethanesulfonyl)imide (LiTFSI) were denoted as PPZ-8, PPZ-16, PPZ-24 and PPZ-40, which corresponded to 8:1, 16:1, 24:1, and 40:1 molar ratios of oxygen atoms in the PPZ to lithium ions in the lithium salt (O:Li⁺), respectively.

Combustion and self-extinguishing time (SET) tests were carried out to verify the fire safety of the materials. A commercial liquid electrolyte (LE, 1 M LiPF₆ in DMC/EMC/EC, v:v:v = 1:1:1) and the liquid polymer electrolyte (LPE, containing PPZ and LiTFSI) were loaded on a

nonflammable glass microfiber membrane (GF). Arc ignition was first employed to simulate a battery spark discharge. As shown in Fig. 2d–e, the continuous arc did not ignite the LPE (GF-PPZ-16). Conversely, the LE immediately caught fire upon ignition. Furthermore, the LE and LPE (GF-PPZ-16) were placed in battery cases and directly exposed to a butane flame. The LPE (GF-PPZ-16) still was not set on fire (Fig. 2f and Supplementary Fig. 1), but the LE ignited, and burning continued even after removing the flame (Fig. 2g). Significantly, the SET of LPE (GF-PPZ-16) was 0 for both ignition modes, which clearly demonstrated the excellent nonflammability of the LPE (Fig. 2h) resulting from the abundant N and P flame retardant elements in the PPZ; these can scavenge the active hydrogen radicals and retard the combustion chain reaction[34,35].

Ion conduction in a polymer-based electrolyte is controlled by the degree of lithium salt dissociation and the coordination environment for Li⁺[19]. Accordingly, Fourier transform infrared (FTIR) spectra and Raman spectra were used to investigate the interactions of the lithium salt with the PPZ. As shown in Fig. 3a and Supplementary Fig. 2a, for the LPEs with increasing lithium salt contents, the FTIR signals for P=N-P (1200–1230 cm⁻¹), C-O-C (1275–1020 cm⁻¹), and P-O-C (970 cm⁻¹) moieties underwent different shape changes and frequency shifts, showing the strong relationships to the lithium salt concentration. It can be inferred that the polar N and O atoms in the PPZ coordinated the Li⁺, which facilitated dissociation of the LiTFSI to form the carriers. After normalizing the Raman spectra of the LPEs with the flexural vibrational signal for CH₂ groups at 1455.7 cm⁻¹, the peak for TFSI⁻ at 780–720 cm⁻¹ revealed the form of the LiTFSI and the relative content (Fig. 3b, Supplementary Fig. 2b and Table 1)[36]. In pure LiTFSI, the Li⁺ and TFSI⁻ formed aggregates (AGG, one TFSI⁻ interacting with two or more Li⁺). For the LPEs, the PPZ dissociated the LiTFSI to produce contact ion pairs (CIP, one TFSI⁻ interacting with one Li⁺) and free TFSI⁻ anions[37,38].

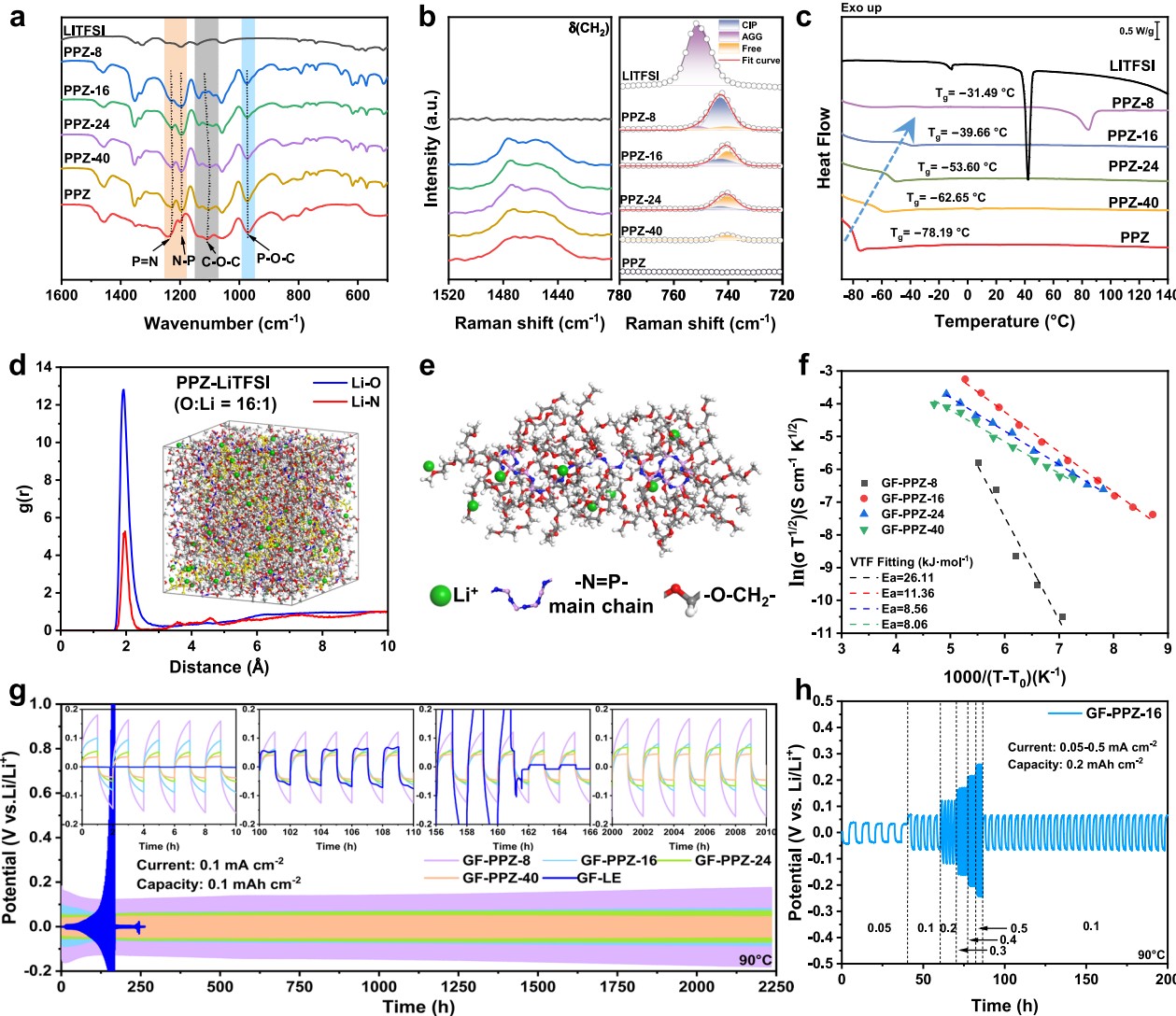

**Fig. 3 | Chemical and electrochemical properties of LPEs. a** FTIR spectra of LPEs. **b** Raman characteristic peaks of TFSI⁻ in LPEs after fitting, the curve was normalized by the signal of CH₂ at 1455.7 cm⁻¹. **c** DSC curves of LPEs. **d** Radial distribution functions g(r) of Li-O and Li-N pairs from MD simulation of PPZ-16, the inset is the MD simulation snapshot. **e** Coordination structure of Li⁺ and PPZ in PPZ-16. **f** Ionic conductivity of GF-LPEs with different Li-salt concentrations as a function of temperature, the dashed lines represent the VTF fitting results. Voltage profiles of

lithium-metal plating/stripping in Li//Li symmetrical cells with (**g**) constant current density of 0.1 mA cm⁻² (capacity: 0.1 mAh cm⁻²) and (**h**) variable current density from 0.05 to 0.5 mA cm⁻² (capacity: 0.2 mAh cm⁻²) at 90 °C. The insets are voltage profiles of Li/LPE/Li cells at 0–10 h, 100–110 h, 156–166 h and 2000–2010 h, respectively. Source data are provided as a Source Data file. Molecular dynamics simulations data are provided as Supplementary Data 1 file.

In particular, when each repeat unit of the PPZ corresponded to a lithium salt, as in PPZ-8, the CIPs were obviously dominant. This special state led to a singular endothermic peak in heating curve at 84.50 °C (Fig. 3c). As the lithium salt content in the LPEs decreased, the free state gradually predominated and exceeded 50%.

Molecular dynamics (MD) simulations were employed to determine the coordination structures and diffusion coefficients of Li⁺ in the LPEs (molecular dynamics simulations data are provided as Supplementary Data 1 file). The radial distribution functions of the Li-O and Li-N pairs were calculated and are displayed in Fig. 3d. The strong peaks for the Li-O pairs at 1.93 Å and the Li-N pairs at 1.97 Å indicated that both the N and O atoms of the LPE were coordinated with Li⁺, which was consistent with the FTIR and Raman results. The details of the coordination structures are shown in Fig. 3e, and the well-distributed Li⁺ ions were transported through the movements of the phosphazene backbone and ether side chains in the PPZ. The diffusion coefficient calculated for Li⁺ in the LPE was 2.5 × 10⁻⁸ cm² s⁻¹ at 298 K, which was slightly lower than that of amorphous PEO (3.2 × 10⁻⁸ cm² s⁻¹). However,

it is commonly known that PEO is semicrystalline below its melting point (65 °C). Ion transport occurs mainly in the amorphous regions of polymer electrolytes and can be enhanced if the polymer has a low glass transition temperature ($T_g$). Differential scanning calorimetry (DSC) measurements were performed to estimate the $T_g$ values of pristine PPZ and the LPEs (Fig. 3c and Supplementary Table 2). Amorphous PPZ showed the lowest $T_g$ of -78.19 °C. Although the lithium salt increased the $T_g$ of the LPE by occupying the free volume and coordinating, the $T_g$ were still much lower than RT. The $T_g$ of PPZ-8, PPZ-16, PPZ-24 and PPZ-40 were -31.49 °C, -39.66 °C, -53.60 °C and -62.65 °C, respectively. As shown in Supplementary Fig. 3, the $T_g$ of PPZ-16 containing GF separator (GF-PPZ-16) was -38.34 °C, and not any thermal transition can be found in DSC curve of the GF, which means that introducing inert GF only caused a trivial change of 1.32 °C.

In addition, the thermal stability of the material determines the upper limit for the application temperatures. As shown in Supplementary Fig. 4 and Supplementary Table 3, PPZ presented nice thermal stability with an initial decomposition temperature (5% weight loss

temperature, $T_{5\%}$) of 280 °C. The $T_{5\%}$ of the PPZ-16 and GF-PPZ-16 were 230.0 °C and 230.5 °C, respectively, indicating that GF had no effect on the initial decomposition temperature of the electrolyte. The reason should be contributed to highly thermal stability of GF at 40–700 °C, with a mass loss of only 2.54%. Compared with PPZ, for PPZ-16 and GF-PPZ-16, the decrease of $T_{5\%}$ might be due to the catalytic decomposition of the side groups by LiTFSI. Even so, $T_{5\%}$ of LPE was still much higher than the operating temperatures of LMBs.

Liquid LPEs do not have the ability to separate the cathode and anode, so a porous separator is still necessary in electrochemical test. Glass microfiber membrane (GF) has intrinsic chemical inertness, good thermal stability, high temperature electrochemical stability, non-flammability and porosity[39]. Moreover, GF hardly affected the thermal properties of LPEs. Therefore, it was considered to be the optimal choice of separator. More details of LPEs impregnating GF were shown in the section of Electrochemical performance of the Li metal batteries.

The LPEs exhibited high electrochemical stabilities with wide electrochemical stability windows (ESW) of up to 5.20 V based on flat plate electrodes (or 4.55 V based on graphite porous electrodes) at 90 °C, which exceeded those of LEs and PEO (Supplementary Fig. 5). Therefore, the LPEs are expected to be suitable for use in high-voltage lithium metal battery systems exhibiting higher energy storage densities. The oxidation potential in polymer electrolyte is mainly determined by the oxidation of polymer host[40], but performing computational works on entire polymer chains becomes prohibitive[41]. The fragment model can reveal the role of monomer chemistry in ESW[42]. Therefore, the first-principles density functional theory (DFT) was used to calculate the redox potential and ESW width (the gap between oxidation potential and reduction potential) of simplified fragment single-chain model of polyphosphazene with different side chain lengths in Supplementary Fig. 6 (DFT calculations data are provided as Supplementary Data 2 file). As the side chain became longer, the oxidation potential increased from 5.94 to 6.09 eV and the reduction potential decreased slightly, the ESW width increased from 7.22 to 7.47 eV. Therefore, the oxidation stability of PPZ can be considered to increase with the increase of side chain length. The illustration in Supplementary Fig. 6 showed that the highest occupied molecular orbital levels were mainly distributed along the N/P main chain, while the lowest unoccupied molecular orbital levels were confined to the terminal. The shielding effect of long ether side chain with high oxidation stability on the lone electron on N atom may be the reason for the improvement of oxidation stability. In addition, it should be emphasized that the real ESW is affected by the electrode and lithium salt[40,41].

Using electrochemical impedance spectroscopy (EIS), the ionic conductivities ($\sigma$) of different LPEs were determined at various temperatures (Supplementary Fig. 7 and Supplementary Table 4). GF-PPZ-40 exhibited the largest ionic conductivity of 0.109 mS cm⁻¹ at 25 °C. However, at 100 °C, the ionic conductivity of GF-PPZ-16 was the largest at approximately 2.01 mS cm⁻¹, reaching the order of magnitude for LEs. The $\sigma$ values of GF-PPZ-8, GF-PPZ-24 and GF-PPZ-40 were 0.158, 1.27, and 0.93 mS cm⁻¹, respectively. As expected, the LPEs exhibited excellent ion transport abilities, especially at RT, which were 1–2 orders of magnitude greater than that of PEO[21,42]. Generally, the $\sigma$ of electrolyte is the product of ionic mobility ($\mu$) and free carrier concentration ($c$)[19]. Ion transport through polymer electrolytes is intimately coupled to the motion of the polymer chain. The activation energy ($E_a$) was calculated with the Vogel–Tamman–Fulcher (VTF) empirical equation. As shown in Fig. 3f and Supplementary Table 4, the $E_a$ values for GF-PPZ-8, GF-PPZ-16, GF-PPZ-24, and GF-PPZ-40 were 26.11, 11.36, 8.56, and 8.06 kJ mol⁻¹, respectively. Good linear fitting relationship showed that ion transport mechanism in LPEs was consistent with that in SPEs. The $E_a$ value reflected the influence of temperature on the ion transport. Lithium salt concentration affected the movement of polymer chain segments to varying degrees, resulting in the difference of activation energy. Since both lithium salt content and temperature have effects on carrier concentration and ion mobility, although it becomes difficult to discuss single variables, the changing trend of ionic conductivity with temperature at different O:Li⁺ ratios was qualitatively analyzed. At RT, GF-PPZ-40 has the highest $\sigma$ due to its lowest $E_a$. With the increase of temperature, the movement ability of polymer chain enhanced, and the $\sigma$ was controlled by both carrier concentration and activation energy. Therefore, at 60 °C, the $\sigma$ of GF-PPZ-24 and GF-PPZ-16 were greatly increased, but the increase of GF-PPZ-40 was limited. At 90 °C, the influence of activation energy became more weaker, while the carrier concentration was dominated, resulting in GF-PPZ-16 showing the largest $\sigma$.

In addition, the lithium-ion transference number ($t_{Li^+}$) showed a positive correlation with the lithium salt content. Specifically, at 90 °C, GF-PPZ-8 reached the largest $t_{Li^+}$ of 0.38, and with a further increase in the O:Li⁺ ratio from 16 to 40, as with PPZ-16 and PPZ-40, the $t_{Li^+}$ decreased from 0.30 to 0.1 (Supplementary Fig. 8). The $t_{Li^+}$ of various GF-PPZ at a range of RT (25 °C) to 90 °C were tested and shown in Supplementary Table 5, indicating no significant temperature dependence. The low $t_{Li^+}$ can be attributed to the "pocket effect"[43,44]. As described in Supplementary Fig. 9 (Molecular dynamics simulation), Li⁺ was bound into PPZ by Li-O and Li-N bonds, while TFSI⁻ anions were distributed around the polymer chain due to lack of interaction and large steric hindrance. Therefore, under an electric field, anions have a faster migration capacity resulting in a low $t_{Li^+}$. Especially in PPZ-40, LiTFSI was completely dissociated by PPZ (Fig. 3b and Supplementary Table 1), meaning that all free Li⁺ was bonding, resulting in the lowest $t_{Li^+}$. With the increase of LiTFSI content, the non-free state Li⁺ (CIP and AGG) was considered to weaken the interaction between Li⁺ and PPZ and thus increased the transference number. The effective Li⁺ conductivity ($\sigma_{Li^+}$) was calculated in Supplementary Table 5 at various temperatures. Due to its good performance in both $\sigma$ and $t_{Li^+}$, GF-PPZ-16 had the highest $\sigma_{Li^+}$ at nearly all temperatures. At 90 °C, the $\sigma_{Li^+}$ of GF-PPZ-16 reached $4.02 \times 10^{-4}$ S cm⁻¹, and the $\sigma_{Li^+}$ of GF-PPZ-8, GF-PPZ-24 and GF-PPZ-40 were $3.15 \times 10^{-5}$, $1.57 \times 10^{-4}$, $8.64 \times 10^{-5}$ S cm⁻¹, respectively.

Interestingly, the LPEs also exhibited excellent inhibition of lithium dendrite formation. The galvanostatic cycling tests demonstrated low Li plating/stripping overpotentials at 90 °C in Li/LPEs/Li symmetrical cells. These cells exhibited ultralong stable cycles over 2200 h, which were only accompanied by very slight fluctuations in voltage (Fig. 3g). The magnification of voltage curves at different times showed that there was no micro-short circuits or voltage drops caused by lithium dendrites[45]. Scanning electron microscopy (SEM) images in Supplementary Fig. 10 showed a dense interface layer on the surface of the lithium metal electrode after cycling, and the cross-section images showed that the deposition of Li⁺ mainly occurred below the interface layer. Energy dispersive spectrometer (EDS) images showed that the elements were evenly distributed in the interface (Supplementary Fig. 11). X-ray photoelectron spectroscopy (XPS) revealed that the interface was composed of organic fragments and inorganic lithium compounds derived from PPZ and LiTFSI (Supplementary Fig. 12), mainly containing LiF, $Li_3N$, $Li_2CO_3$, $Li_3PO_4$, etc. This dense hybrid interface ensured good transport of Li⁺ and impeded the growth of lithium dendrites. Conversely, the cell with the LE showed rapidly increasing overpotentials (over 1.8 V after 160 h) and finally failed within only 170 h. Furthermore, the Li/GF-PPZ-16/Li cell was tested at current densities varying from 0.05 to 0.5 mA cm⁻², and the cell still operated stably with acceptable voltage increases (Fig. 3h). To rule out dendrite growth inhibition by the high-strength GF separator, the plating/stripping voltage profiles of the cells based on the cellulose membrane (CM) separator were also tested. Similar results were obtained in Supplementary Fig. 13. Overall, the effective suppression of Li dendrite growth by the LPEs will contribute substantially to the excellent performance of the Li metal battery.

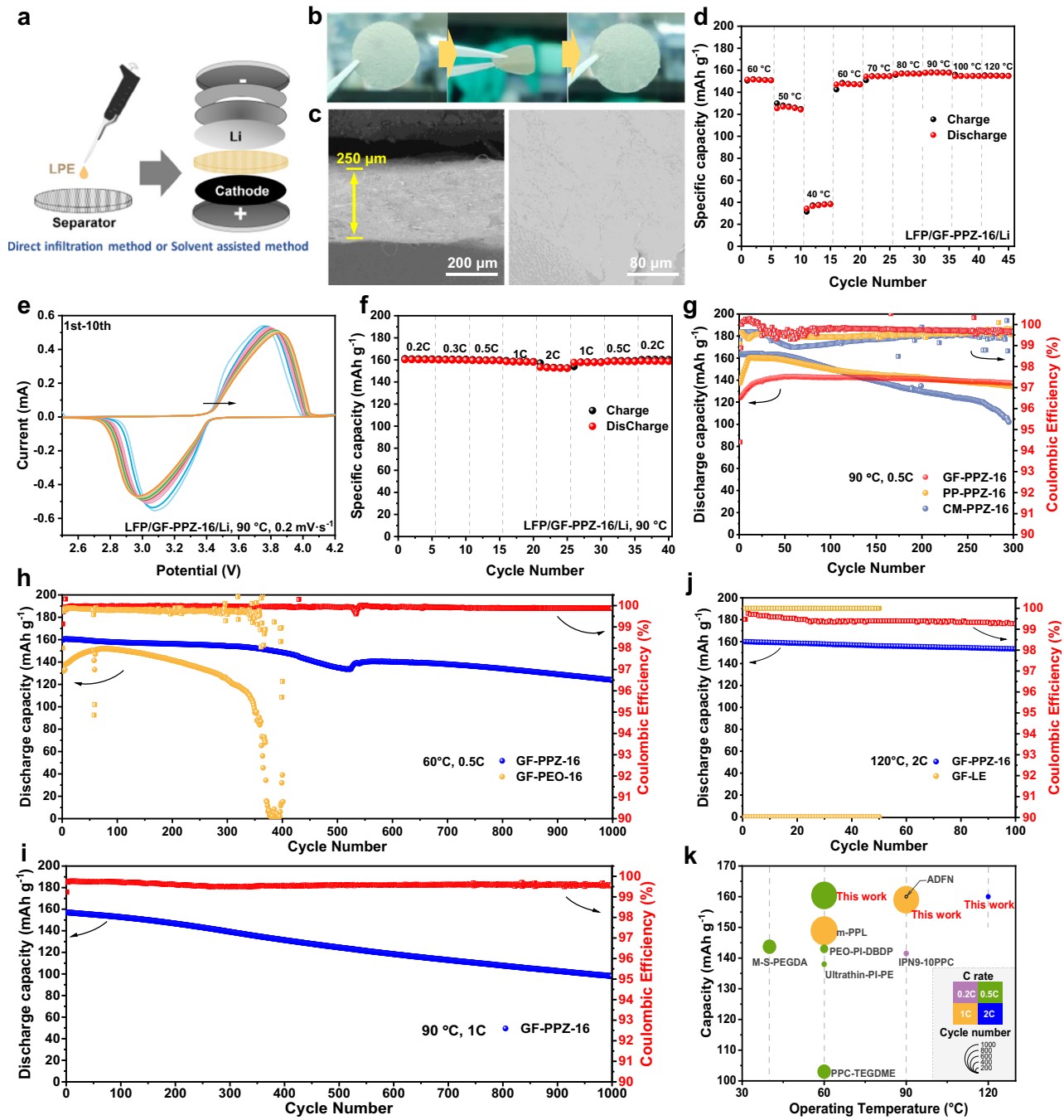

**Fig. 4 | Fabrication and performance of LPE-based cells. a** Schematic illustration of LPE-based LMBs' preparation. **b** Digital pictures of GF-PPZ−16, 180°-fold and unfold. **c** Cross-sectional and surface SEM images of GF-PPZ−16. **d** Capacity performance of LFP/GF-PPZ−16/Li cell at 0.5C with various temperatures. **e** CV curves of LFP/GF-PPZ−16/Li cell with a scan rate of 0.2 mV s⁻¹. **f** Rate performance of LFP/ GF-PPZ-16/Li cell at 90 °C. **g** Cycling performance of LPE-based LMBs with different separators. Long-cycle performances of LPE-based LMBs (**h**) at 60 °C under 0.5C, (**i**) at 90 °C under 1C, and (**j**) at 120 °C under 2C. **k** Comparison of cycle performance (Corresponding references can be found in Supplementary Table 6). Source data are provided as a Source Data file.

## Electrochemical performance of the Li metal batteries

To assess the feasibility of using the LPEs in practical batteries, the LPEs were loaded on GF separators to assemble Li metal full batteries containing a LiFePO₄ (LFP) cathode (Fig. 4a). Supplementary Fig. 14 showed that under heating at 80 °C, LPE (PPZ-40) can infiltrate gradually to GF separator for 20 min and spread out fully in GF after 5 h. Therefore, the relatively high viscosity made the direct use of LPE for battery assembly inefficient. In order to ensure the uniformity and consistency of electrolytes in separators, LPEs were loaded on separators by a solvent-assisted method (Supplementary Fig. 15). The

solvent (ethylene glycol dimethyl ether) with a low boiling point can be completely removed by vacuum heating. The GF separator filled with the LPE, denoted as GF-LPE, showed good flexibility and completely returned to its original state after 180° folding (GF-PPZ-16, Fig. 4b). Specifically, SEM and EDS images showed that PPZ-16 was uniformly filled into GF separator to form a dense and smooth structure (Fig. 4c and Supplementary Fig. 16). Similar results were found for cellulose membrane (CM) and polypropylene (PP) separators (Supplementary Figs. 17, 18), which indicated that the LPEs were suitable for various commercial separators. In addition, a heating experiment

demonstrates that GF-LPE has good dimensional stability and there is no leakage of LPE at 25–180 °C (Supplementary Fig. 19).

The electrochemical performance of an LPE-based Li metal full battery was investigated. First, the charge-discharge capacities of LFP//Li coin cells with various LPEs and a GF separator, i.e., LFP/GF-PPZ-16/Li, LFP/GF-PPZ-24/Li, and LFP/GF-PPZ-40/Li, were determined at different temperatures (Fig. 4d and Supplementary Fig. 20). All of the LPE-based cells exhibited excellent capacity reversibility between 30 and 120 °C at a current density of 0.5C. In comparison, the LFP/GF-PPZ-16/Li cell delivered the highest capacities at temperatures ranging from 50 to 120 °C, i.e., 128, 152 and 158 mAh g$^{-1}$ at 50, 60 and 90 °C, respectively. In addition, the LFP/GF-PPZ-24/Li cell also presented satisfactory capacities of ~48 and ~95 mAh g$^{-1}$ at 30 and 40 °C, respectively. Overall, the LPE-based Li metal batteries worked well at various temperatures ranging from ambient to high.

The reversibility and activity of the electrochemical reactions occurring in the different LPE-based LFP//Li cells were evaluated by cyclic voltammetry (CV) (Fig. 4e and Supplementary Fig. 21). The oxidation and reduction peak potentials of all cells were approximately 3.75 and 3.08 V, respectively, which corresponded to delithiation and lithiation of the cathode. With more cycles, the peak potentials appeared a certain shift, which may be related to the interface construction and polarization. However, the peak potential offset decreased gradually, indicating a stable interface has been formed. Moreover, the LFP/GF-PPZ-16/Li cell exhibited excellent rate performance with large and reversible capacities of 160.6, 160.3, 160.0, 159.4, 158.2 and 152.6 mAh g$^{-1}$ for current densities of 0.1C, 0.2C, 0.3C, 0.5C, 1C and 2C at 90 °C, respectively (Fig. 4f). The corresponding specific capacity-voltage curves also showed typical charge-discharge profiles between 2.50 and 4.20 V (Supplementary Fig. 22). The overpotential was about 93 mV at 0.2C, reflecting good interface contact and fast kinetics of Li$^+$ transport[46,47], corresponding to low ohmic polarization and activation polarization. As the current density increased, the concentration polarization and activation polarization became apparent resulting in the overpotential increasing to 377 mV at 2C, but still within the acceptable range. The low interfacial polarization was attributed to the good wettability of the liquid state LPE to the electrodes, as well as the stable electrode/electrolyte interface derived from LPE.

The LPE-based LFP//Li cells with different separators or lithium salts were cycled at 90 °C and at 0.5C for long times, and they exhibited average coulombic efficiencies (CE) of approximately 100% (Fig. 4g and Supplementary Fig. 23); this confirms the broad applicability of the LPEs. However, after comprehensively comparing both the capacities and stabilities, the LFP/GF-PPZ-16/Li cell with the GF separator and LiTFSI lithium salt stood out. As expected, ultralong stable cycling times with very high initial capacities and average CEs were achieved. Specifically, the LFP/GF-PPZ-16/Li cell delivered initial discharge specific capacity of up to 160 mAh g$^{-1}$ and a retained discharge capacity of 124 mAh g$^{-1}$ with an average CE of over 99.9% for 1000 cycles run at 0.5C at 60 °C (Fig. 4h). In contrast, the PEO-based LFP/GF-PEO-16/Li cell showed significant capacity decay after 300 cycles.

These LBs can be charged and discharged normally at high temperatures, and this is of great significance in expanding their applicability and ensuring their safety during service[48]. Due to the excellent thermal stabilities and electrochemical properties of the LPEs, an initial discharge specific capacity of 157 mAh g$^{-1}$, a retained discharge capacity of 98 mAh g$^{-1}$ after 1000 cycles and an average CE above 99.6% were obtained for the LFP/GF-PPZ-16/Li cell operated at 1C at 90 °C (Fig. 4i and Supplementary Fig. 24). More amazingly, the LFP/GF-PPZ-16/Li cell showed a high initial capacity of 160 mAh g$^{-1}$ and excellent capacity retention of 95.9% with an average CE above 99.2% over 100 cycles run at 2C and 120 °C, while the cell containing the liquid

electrolyte (i.e., LFP/GF-LE/Li cell) failed completely (Fig. 4j). In addition, the LPEs are highly compatible with high-voltage cathodes even at a high temperature. As shown in Supplementary Fig. 25, two LPE-based LMBs with high-nickel NCM811 (LiNi$_{0.8}$Co$_{0.1}$Mn$_{0.1}$O$_2$) cathodes, i.e., the NCM811/GF-PPZ-24/Li and NCM811/GF-PPZ-16/Li cells, were cycled steadily over 100 times at 0.5C and showed an average CE of approximately 100% at 60 °C.

Overall, compared to the state-of-the-art SPE and LE electrolytes, the LPEs endowed the Li metal batteries with unprecedented advantages in terms of capacity, rate, operating temperature and lifetime (Fig. 4k and Supplementary Table 6). Moreover, the LPEs also presented good comprehensive performance (Supplementary Fig. 26).

## Morphology and chemistry at the interphase

Undoubtedly, the excellent electrochemical performance of the LPE-based Li metal batteries requires a stable interface and good interfacial contact. Therefore, an interfacial adhesion test was used to quantitatively describe the strength of the interfacial contact[49]. The adhesion energies of GP-PPZ-16 sandwiched between the LFP cathode and Cu anode or graphite anode reached 183.80 ± 24.42 and 106.03 ± 33.63 J m$^{-2}$ (Supplementary Fig. 27), respectively. The cell was disassembled after 1000 cycles to observe the morphology of the interface between the LPE and the electrodes. Supplementary Fig. 28 shows that the LPE infiltrated into the interior of the pristine LFP cathode to improve the contact of the electrolyte-cathode interface. On the other hand, the abundant O and N atoms in the PPZ provided good affinity for the Li metal. As a result, a dense and intact interfacial layer was observed on the Li metal (Fig. 5a, b). The thickness of the Li deposit was approximately 45 μm and showed no lithium dendrites. After the polymer component was completely washed off the anode surface, a flat and dense surface was exposed (Fig. 5d), which was similar to the pristine Li metal anode (Fig. 5c).

In addition, XPS spectra determined the chemical composition at the interphase. All peaks in the C $1s$ XPS spectrum were theoretically derived from segments or decomposition products of the LiTFSI or the ether side chains of the PPZ (Fig. 5e). The peaks in the F $1s$ spectrum at 688.6 eV and the N $1s$ spectrum at 399.4 eV corresponded to LiTFSI (Fig. 5f). In addition, the peaks of the F $1s$ spectrum at 684.6 eV, the N $1s$ spectrum at 397.6 eV, and the P $2p$ spectrum at 133.5 eV corresponded to LiF, Li$_3$N, and Li$_3$PO$_4$, respectively (Fig. 5f–h). Because LiTFSI does not produce Li$_3$N in the PEO/LiTFSI SPE[46,50], all Li$_3$N and Li$_3$PO$_4$ must have originated from the PPZ. An important reason for thermal failure at a SEI is decomposition and dissolution of the metastable components, which leads to continuous reactions of the electrolytes with the lithium anode and thickening of the SEI[51]. LiF, Li$_3$N and Li$_3$PO$_4$ all have high melting points, good thermal stabilities and excellent mechanical strengths and are electrical insulators[52]. Therefore, the electrolyte-electrode interface based on the LPEs suppresses lithium dendrite formation and persistent side reactions with a physical barrier, which is crucial for a stable high-temperature interface[53]. On the other hand, Li$_3$N and Li$_3$PO$_4$ are fast Li$^+$ conductors that can effectively tune the Li$^+$ distribution and avoid lithium dendrite formation[54]. The good wettability of the Li anode by Li$_3$N can integrate the interface and reduce the interfacial resistance[55]. It can be concluded that the abundant LiF, Li$_3$N, and Li$_3$PO$_4$ from the LPE endowed the interface with excellent thermal stability, mechanical strength, ionic conductivity, and flexibility, which contributed to the long-term cycling stabilities of the LMBs even under high temperatures.

The Li metal anode exhibits significant volume changes during charge-discharge cycling, which requires that the electrolytes must be flowable or deformable to maintain good contact and integrity at the interfacial layer. Poor interfacial contact and the generated defects can lead to nonuniform charge distributions, destruction and thickening of the interfacial layer, and Li dendrite growth, which are the main reasons for the loss of battery life[56,57]. The limited deformability of a

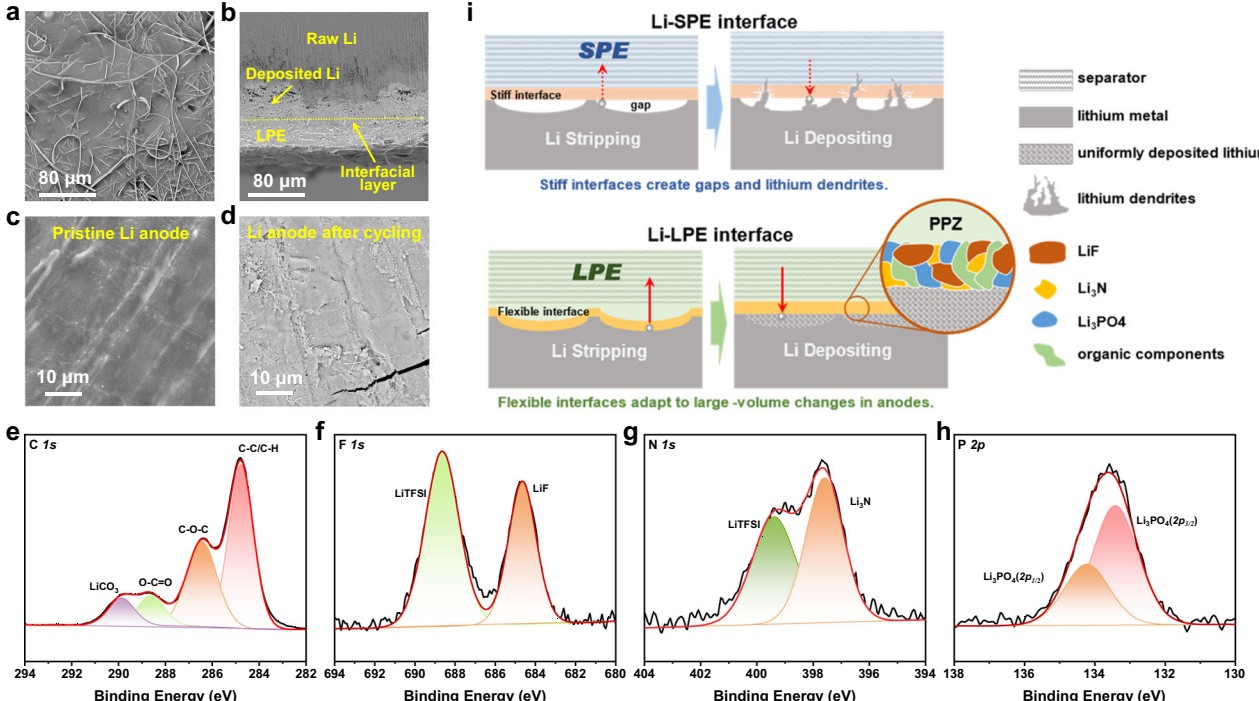

**Fig. 5 | Electrode-electrolyte interface analysis of the LPE-based LMBs.** SEM images of (**a**) surface and (**b**) cross-section of Li anode without removing LPE after 1000 cycles, (**c**) surfaces of pristine Li anode, and (**d**) LPE-removed Li anode after 1000 cycles. XPS spectra of LPE/Li interface, (**e**) C 1s, (**f**) F 1s, (**g**) N 1s, (**h**) P 2p. **i** Schematic illustration of interfacial evolution of LPEs and SPEs during cycling. Source data are provided as a Source Data file.

SPE cannot address these problems[27]. In contrast, the fluidity of the LPE allows it to wet Li metal anodes like a LE at any time to form complete and continuously flexible interfacial layers consisting of inorganic and organic composite components (Fig. 5i). Moreover, the unimpeded charge transfer channel enables uniform plating and stripping of lithium below the interfacial layer.

## All-around safety of Li metal batteries

Due to the good thermotolerance and superior flame retardance of the LPEs, an accelerating rate calorimeter (ARC) was used to study the thermal runaway behavior of LPE-based and LE-based coin cells (Fig. 6a). The temperatures corresponding to heating rates of 0.02 °C min⁻¹ and 10 °C min⁻¹ are defined as the onset temperature for exothermic reactions ($T_{OER}$) and the thermal runaway temperature ($T_{TR}$), respectively[34]. The $T_{OER}$ of the LPE-based cell was 190.5 °C, which was 50 °C higher than that of the LE-based cell. This indicated that the LPE significantly suppressed the exothermic reactions and delayed the thermal destruction. Significantly, for the LPE-based cell, the maximum heating rate was only 0.49 °C min⁻¹, and then no $T_{TR}$ was detected, which meant that no thermal runaway occurred throughout the ARC test. In contrast, the LE-based cell reached a heating rate of 10 °C min⁻¹ at 414.5 °C, and unsurprisingly, thermal runaway occurred.

To verify the practical commercial application potential of the LPEs, pouch cells were fabricated for testing cycling performance and safety in harsh environments. As shown in Fig. 6b and Supplementary Fig. 29, the discharge specific capacity of the LFP/LPE(CM-PPZ-16)/Li pouch cell was greater than 140 mAh g⁻¹ when tested at 0.2C and 90 °C, and its average coulombic efficiency exceeded 99.2%. It is foreseeable that LMBs used in high-temperature environments will face more serious safety risks, including drastic changes in the external temperature and internal heat accumulation. To verify that the LPE-based LMBs were safer, a simple experimental setup was constructed to test the safety of the pouch cell during thermal abuse (Fig. 6c and Supplementary Fig. 30). The red LED was continuously lit from 30 to 210 °C with a stable operating voltage (Supplementary Fig. 31), and

there was no expansion or deformation of the cell during the test. Unprecedentedly, the pouch cell was placed in a vacuum oven at 90 °C to simulate the expansion that may occur in a high temperature environment or at a high altitude. For the LE-based pouch cell, the low-boiling liquid electrolyte caused the battery to swell significantly and fail under vacuum conditions (Supplementary Fig. 32). In contrast, due to the excellent stability of the LPE formed without any small-molecule solvents or plasticizers, the LPE-based pouch cell worked for more than 1 h at 90 °C and 1.3 kPa and was still intact in appearance after the test (Fig. 6d).

To further highlight the fire safety of the LPE-based LMBs, a combustion test was conducted with the pouch cells. The LE-based pouch cell was rapidly ignited within 0.5 s and continued to burn for 15 s with an intense flame (Fig. 6e). However, the LPE-based pouch cell was not be ignited even after 5 s (Fig. 6f), demonstrating that the nonflammable LPEs significantly improved the fire safety of the LMBs.

In addition, the LPE-based pouch cell exhibited excellent flexibility and high resistance to mechanical abuse. During folding, penetration or cutting, the cell continued to light the LED (Fig. 6g–i). These results indicate that the LPE will endow practical LMBs with remarkable performance and very high safety under various harsh environments or in the case of abuse.

## Discussion

In summary, a nonflammable solvent-free LPE was developed for high-performance and safe Li metal batteries. Due to a room-temperature liquid-state brush-like polymer consisting of a polyphosphazene backbone and oligomeric ethylene oxide side chains, the LPE simultaneously addressed the most pressing challenges faced by LMBs containing LEs or SPEs, such as electrolyte oxidation on cathode, Li-dendrite formation, low thermotolerance, poor Li plating/stripping, interfacial instability, poor safety, and manufacturing difficulty. Our electrolyte delivers high performance over a wide temperature range (60–120 °C) and enables Li plating/stripping for more than 2200 h as well as stable cycling of LFP//Li and NCM811//Li with high coulombic efficiency and capacity retention.

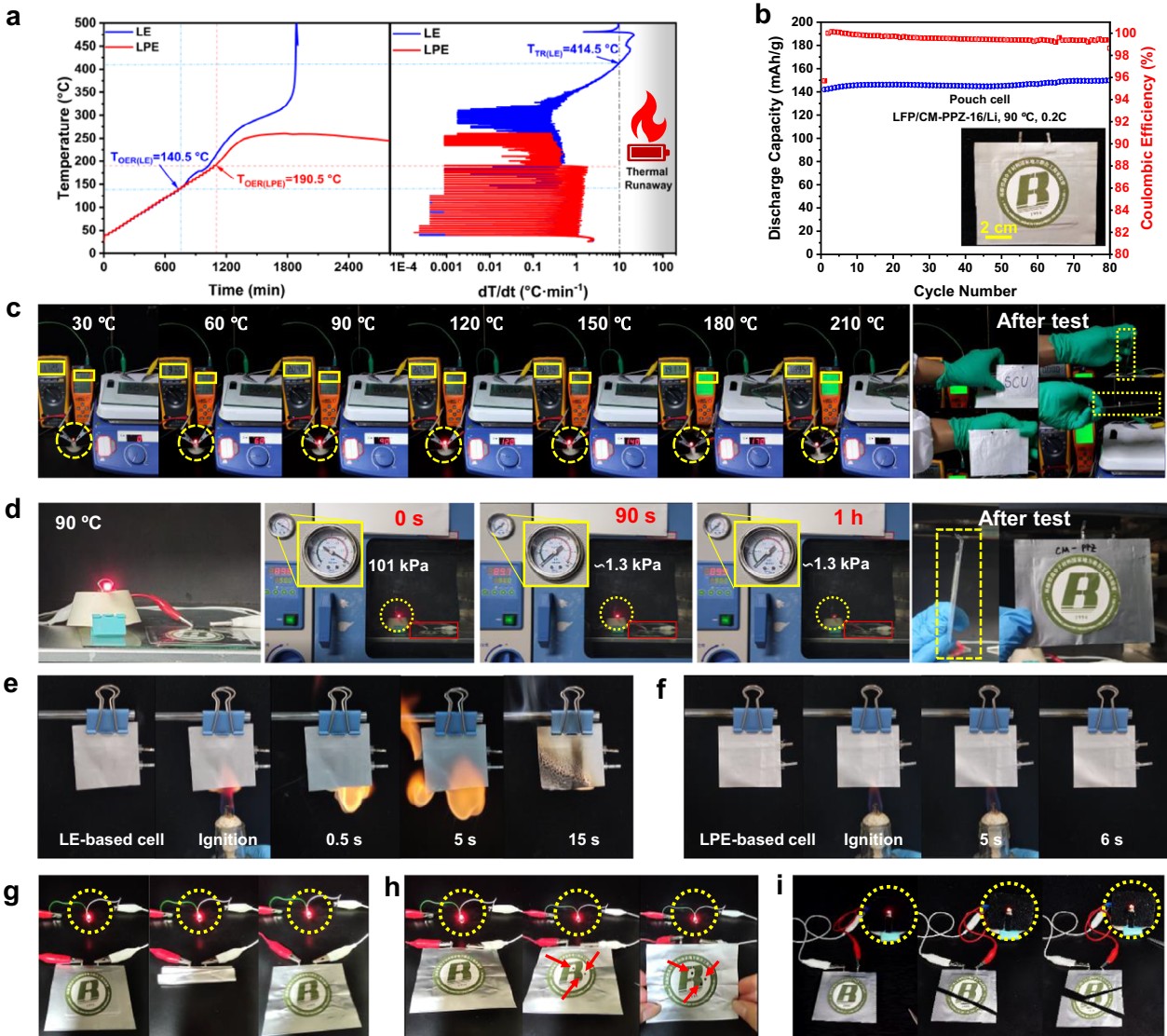

**Fig. 6 | Comprehensive safety testing of LPE-based LMBs. a** Temperature vs. time curves of LFP/LE/Li and LFP/LPE/Li coin cell in ARC test with heat-wait-seek (HWS) mode. **b** Cycling performance of LPE-based (LFP/CM-PPZ-16/Li) pouch cell at 90 °C, and inset is the digital photo of the pouch cell. Safety of cells in extreme environments, (**c**) thermal abuse test on continuous heating flat heater, (**d**) vacuum-heating test in vacuum environment of 90 °C, and combustion experiment of pouch cell with (**e**) LE-based electrolyte and (**f**) LPE-based electrolyte. Digital photographs of pouch cells operating under harsh mechanical abuse, (**g**) multiple folding, (**h**) penetration, and (**i**) cutting. Source data are provided as a Source Data file.

More importantly, the LPE-based LMBs have the capacity to suppress thermal runaway and resist mechanical abuse. The design concept of this nonflammable LPE without any small molecular solvents provides innovative insights for developing next-generation safe and high-performance rechargeable batteries.

## Methods
### Materials
Phosphonitrilic chloride trimer (HCCP), triethylene glycol monomethyl ether (TGME), sodium hydride (NaH), lithium bis(trifluoromethanesulphonyl)imide (LiTFSI), and ethylene glycol dimethyl ether (DME) were all acquired from Aladdin Biochemical Technology Co., Ltd. (Shanghai, China). LiFePO₄ (LFP), Super P Li and Poly(vinylidene fluoride) (PVDF), NKKT4030 Cellulose membrane and Celgard2500 PP separator were provided by Canrd New Energy Technology Co., Ltd. (Dongguan, China). Glass microfiber membrane (GF) were offered by Whatman™. Tetrahydrofuran (THF), petroleum ether and N-methyl pyrrolidone (NMP) were

provided by KeLong Co., Ltd. (Chengdu, China). All the materials were used without any further purification.

### Synthesis of PPZ
Poly bis-(methoxytriethoxy) phosphazene (PPZ) was prepared via a phosphonitrilic chloride trimer (HCCP) melt bulk polymerization process[58]. HCCP was sealed into a vacuum glass tube and heat it at 260 °C for 36 h to obtain a translucent colloidal primitive poly(dichloro) phosphazene (PDCP) polymer. Then, PDCP was dissolved in ultradry THF and precipitated with dry petroleum ether to remove unreacted monomers. TGME reacted with NaH to obtain sodium alkoxide. After all of this, the PDCP solution was added dropwise into the sodium alkoxide solution and kept stirring for 12 h at 0 °C and then the temperature was heated to 60 °C for 12 h. After solvent evaporation, the original PPZ was purified in a dialysis tube (molecular weight cut off 14,000 Da) against distilled water. The final brown viscoelastic glue PPZ was obtained after removing the water and drying at 80 °C under a vacuum for 48 h. ³¹P NMR: -8.43 ppm. ¹H NMR: 3.36 ppm (s, 3H, OCH₃),

3.61 ppm (t, 8H, CH$_2$CH$_2$OCH$_2$CH$_2$OMe), 3.53 ppm (m, 2H, CH$_2$O), 4.03 ppm (t, 2H, POCH$_2$).

## Preparation of LPEs and GF-LPEs

LPEs were prepared by mixing PPZ and lithium salt (O: Li$^+$ = 8:1, 16:1, 24:1, and 40:1, molar ratio). For practical use, LPEs were filled into separators (GF, CM, or PP). 75 wt% DME solvent was used to further reduce the viscosity of LPEs to achieve adequate fluidity. Then, the mixed solution (600 μL) was dropped into the GF separator (φ19 mm). Similarly, the mixed solution (100 μL) was added to the PP separator or CM separator. After the solvent evaporated in the air, LPEs-loaded separators were heated to 100 °C for 12 h to make LPEs evenly filled. Then, all LPEs-loaded separators were vacuum dried at 80 °C for more than 48 h to ensure that solvent was removed fully. The load of LPEs in GF, CM, and PP were 121.00 ± 0.22 mg, 20.78 ± 0.04 mg, and 19.25 ± 0.06 mg, respectively. The completely dry electrolyte membranes were quickly transferred into an Ar-filled glove box (H$_2$O and O$_2$ below 0.1 ppm).

## Characterizations of LPEs

The molecular structure of PPZ was characterized by NMR (Bruker AVANCE AV II–400 NMR spectrometer, Bruker, Germany). The molecular weight was determined by GPC (HLC-8320GPC, TOSOH, Japan), the mobile phase was DMF. The chemical structure of PPZ and LPEs was characterized by Fourier transforms infrared (FTIR, Nicolet 6700, US), and Raman (DXR2xi Raman Imaging Microscope, Thermo Fisher Scientific, US). Morphological investigation was characterized by scanning electron microscope (SEM, Phenom ProX). Rheological behavior was determined by a dynamic rheometer (Discovery HR-2, TA, US). TGA (NETZSCH TG 209 F1) was used to analyze the thermal stability of LPEs under Nitrogen at a heating rate of 10 °C min$^{-1}$ from 40 to 700 °C. DSC (TA Q200, US) determined the glass transition temperature and crystallinity at a temperature range from −85 to 140 °C with a rate of 5 °C min$^{-1}$. The adhesion test was carried out according to a reported method with minor variations[49,59]. The sample consisted of LFP-cathode, GF-PPZ-16 electrolyte, and Cu anode (or self-made graphite anode). Cathode and anode were cut into 20 × 80 mm rectangles and GF-PPZ-16 was cut into 20 × 15 mm rectangles. Then align the three along one end and fit them together. The complete sandwich structure was placed at 90 °C for 2 h to fully bond the battery structure. Raw data was measured by an electronic universal testing instrument (INSTRON F563-44, INSTRON Co., Ltd., USA) at RT. Error bars, where present, are calculated from the standard deviation of multiple measurements (>3).

## Electrochemical measurements

All electrochemical measurements were performed on an electrochemical workstation (Princeton ParSTAT). The voltage window was determined by linear sweep voltammetry (LSV) at a scan rate of 1 mV s$^{-1}$ from 0 to 6 V (vs. Li/Li$^+$). The ionic conductivities from 25 to 100 °C were measured by EIS at a frequency range of 0.1M Hz–0.1 Hz in SS/LPE/SS (stainless steel) symmetrical battery. Activation energy ($E_a$) were further calculated by a Vogel–Tamman–Fulcher (VTF) empirical equation[60]: $\sigma = \sigma_0 T^{-1/2} \exp\left(-\frac{E_a}{R(T-T_0)}\right)$, where $\sigma_0$ is the pre-exponential factor, $E_a$ is the activation energy, $T_0$ is ideal glass transition temperature (Here, $T_0 = T_g$–50 °C, $T_g$ obtained from DSC), R is ideal gas constant. The lithium-ion transference number ($t_{Li}^+$) was obtained by the chronoamperometry test in Li/LPE/Li symmetrical cell. The AC impedance plots were measured from 0.1M Hz to 0.1 Hz. Effective lithium-ion conductivities ($\sigma_{Li^+}$) were deduced by equation: $\sigma_{Li^+} = \sigma \cdot t_{Li^+}$. Galvanostatic cycling measurement was performed on Li/LPE/Li cells with a current density of 0.1 mA cm$^{-2}$, wherein the plating/stripping cycle was 2 h.

## Molecular dynamics simulations

All the calculations were performed in Material Studio 2019. First, a periodic simulation box with fixed amounts of PPZ and LiTFSI (O: Li = 16:1) was created by an amorphous cell. Next, the unit cell was optimized by the energy minimization and MD simulation in the NPT ensemble. MD simulation was carried out at 298 K by application of the Velocity Scaling thermostat. The total simulation time was 2000 ps, in the NVT ensemble, the time step was 1 fs, and the trajectory was recorded every 2000 fs. Electrostatic interaction and van der Waals interaction adopted Ewald summation and Atom-based summation, respectively, with a calculation accuracy of $1 \times 10^{-5}$ kcal mol$^{-1}$ and a spline cutoff distance of 18.5 Å. The COMPASS II force field was used for the whole process. Finally, the simulation results' mean square displacement (MSD) was analyzed by Forcite module. And the diffusion coefficient (D) of lithium ions was calculated from MSD by equation: $D = \lim_{t \to \infty} \frac{1}{6t} <|r(t) - r(0)|^2>$.

## Density functional theory (DFT) calculations

DFT calculations were performed on ORCA 5.0.3 program[61] based on a simplified fragment chain model of PPZ[58]. All structures were optimized in the B3LYP hybrid exchange-correlation functional as well as 6-31 G* basis sets[62]. The vdW interactions were described using DFT-D3 (BJ) dispersion correction. Moreover, The Visual Molecular Dynamics (VMD) and Multiwfn software were used for analysis and mapping[63], isovalue = 0.03.

## Manufacture and performance measurements of coin and pouch cells

LFP cathodes contain 80 wt% LiFePO$_4$, 10 wt% Super P, 10 wt% PVDF. The active material load was about 1.6 mg cm$^{-2}$. LFP cathode was cut into 12 mm diameter disks for the CR2032 type coin cell and 5 × 5 cm square pieces for the pouch cell. All types of batteries were assembled in an argon-filled glove box with less than 0.1 ppm O$_2$ and 0.1 ppm H$_2$O. Long-term cycling, rate capability, and temperature-capability performance of LFP/LPE/Li batteries were measured in a high-temperature environmental chamber (BPH-060A, Shanghai Yiheng Scientific Instrument Co., Ltd.) by a battery test system (Shenzhen Neware Electronics Co., Ltd) between 2.5 V and 4.2 V. Environmental temperatures include 60, 90, and 120 °C. Safety testing based on fully charged LFP/LPE/Li batteries (4.2 V, 100% state of charge (SOC)). The thermal runaway behavior of coin cells was obtained by an accelerating rate calorimeter (ARC, BAC-90A, Hangzhou Young instruments Technology Co., Ltd., China) with a heat-wait-seek mode (HWS: tart temperature was 40 °C, heating rate was 5 °C min$^{-1}$, step temperature was 5 °C, time-keeping was 30 min, monitoring threshold was 0.02 °C min$^{-1}$). The capacity of coin cells was about 0.27 mAh. The thermal abuse of pouch cell was carried out on a self-made device (Supplementary Fig. 30). Vacuum-heating test was carried out in a vacuum oven (DZF-6050, Shanghai Yiheng Scientific Instrument Co., Ltd.).

## Data availability

All data supporting the findings of this study are available within the article, as well as the Supplementary Information file, or available from the corresponding authors upon reasonable request. Source data are provided with this paper.

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

## Acknowledgements

This work was supported by National Natural Science Foundation of China (51773134), the Sichuan Science and Technology Program (2019YFH0112), the Fundamental Research Funds for the Central Universities, Institutional Research Fund from Sichuan University (2021SCUNL201), and the 111 Project (B20001). We also thank the Analysis and Testing Center of Sichuan University for NMR testing.

## Author contributions

G.W. and Y.Z.W. conceived and directed the research. G.R.Z. designed and performed most of the experiments with the help from the other authors. Q.Z. participated in parts of the experiments. Q.S.L. participated in the molecular dynamics simulations and DFT calculations. Q.Y.B., Y.Z.Q. and Y.G. assisted with experiments. G.R.Z. wrote the initial manuscript draft. G.W. edited the draft. All authors contributed to the revised manuscript.

## Competing interests

The authors declare no competing interests.
