## [Peer Review File · Nature Communications]

Non-flammable solvent-free liquid polymer electrolyte for lithium metal batteriesREVIEWER COMMENTS

Reviewer #1 (Remarks to the Author):

This paper reports non-flammable liquid electrolytes for applications to Li-metal batteries. Overall, the characterization and practical application of liquid-state electrolytes are systematically investigated, and the results look promising. I think it should be published after minor revisions, as listed below.

- 1) It seems LPE exhibited a wide electrochemical window up to 5.2 V, potentially benefiting the application into the battery cells with high potential cathodes. The authors demonstrated the feasible implementation of LPE into battery cells with high-potential cathodes, but it shows significant capacity degradation over the cycle even though the loss in Coulombic efficiency and interfacial issue are negligible. I think the physic underlying such a high potential window is not thoroughly studied in this paper, likely helping to understand the capacity degradation. Further, even though the electrolytes have a high potential window, it does not indicate the stability in high potential batteries as there is an interfacial issue between the cathode and electrolyte. I think the use of porous electrodes for an electrochemical window test would help to mimic the actual situation.
- 2) Even though it is a liquid electrolyte, the ionic conductivity at RT is very low. Please rationale it. This drives to very low capacity in the battery cells compared to a theoretical value at 30 and 40 degree Celsius.
- 3) Please calculate the charge passed through from the Galvanostatic cycling, which would help to compare the result with the literature value.
- 4) Please explain the reason for weak interfacial polarization in LPE-electrode interface (charge and discharge curve).
- 5) The battery cells showed very stable cyclability at 120 degree Celsius, but the significant capacity degradation was observed at 60 and 90 degree Celsius even though the loss in Coulombic efficiency and interfacial issue are negligible. Please rationale it.
- 6) Please provide the adhesion energy of GP for comparison.

Reviewer #2 (rephrased by editor)

The manuscript entitled "Non-flammable solvent-free liquid polymer electrolyte for lithium metal batteries" reports on stable long-term cycling of lithium metal batteries at high temperatures (60 and 90°C) using a solvent-free liquid poly[bis-(methoxy)triethoxy phosphazene] (PPZ) with non-flammability and high thermal stability. However, the current version of the manuscript contains only limited data and lacks a comprehensive discussion. Therefore, the authors need to provide more data and explanations to optimize the amount of salt and improve the electrochemical performance. In addition, the organization of data is inconsistent among conductivity, electrochemical stability, and cell performance. Based on these issues, it is not recommended for publication in the current form. The following issues need to be addressed:

1 The filling of PPZ into GF separator needs to be demonstrated more clearly. This includes more experiments for filling PPZ into GF separator, explanation of how to fill GF separator with PPZ given its viscosity, what kind of solvent was used for this preparation, and evidence such as SEM-EDS mapping and contact angle between PPZ and GF separator in Fig. S8. Porosity before and after filling PPZ into GF separator should be investigated, and the filling percentage (or PPZ uptake) should be determined. The authors should also show data to prove whether there is any leakage of PPZ after filling.

2 In LSV investigation, the authors need to provide more detailed explanations of why the GF-PPZ-16 (O:Li⁺ ratio of 16:1) has much higher electrochemical stability (5.2 V) when compared to the GF-LE (4.2 V). The full-profile (containing lower potential limit) of LSV

curves should be provided to evaluate the Li plating/stripping behavior.

3 The authors need to discuss the mechanism of Li-ion transport more clearly, including the effect of O:Li⁺ ratio on ionic conductivity in more detail at RT, 60, 90, and 100°C. The increasing trend of ionic conductivity was not consistent at RT, 60, and 90°C. The highest ionic conductivity at RT, 60, and 90°C was obtained from GF-PPZ-40, GF-PPZ-24, and GF-PPZ-16, respectively (see Table S4). The authors need to explain the reason for this difference in the highest ionic conductivity vs. temperature at various O:Li⁺ ratios.

4 In Li-ion transference number (t_{Li^+}) analysis, the authors need to explain why t_{Li^+} increased from 0.1 to 0.3 with increasing O:Li⁺ ratios from 16 to 40. The t_{Li^+} of GF-LE should be shown and compared with GF-PPZ electrolytes. Additionally, the authors should explain why they conducted t_{Li^+} tests at 90°C while other properties (LSV, cell performance) were measured at RT and 60°C. The t_{Li^+} of GF-PPZ and GF-LE should be provided at RT and 60°C.

5 The authors need to show all data of the effective Li-ion conductivity at various temperatures fully along with more discussion to prove that the GF-PPZ-16 has the highest effective Li-ion conductivity.

6 The authors need to clarify why Li-dendrite growth was suppressed in the Li/PPZ-16/Li cell. Some evidence should be provided, such as SEM and XPS analysis of Li metal before and after 2500 h cycling.

7 Why is GF-PPZ-16 light yellow in color in Fig. S8 while others are white? The scale bar in the cross-sectional SEM images is not consistent between Fig. S8 and Fig. S9. It is not convincing to claim that the GF separator is the best choice among GF, CM, and PP separators solely based on SEM images showing its dense and smooth structure. More information on the effective Li-ion conductivity of the separators fully dipped in PPZ-16 should be provided for a fair comparison.

8 It is not possible to conclude that the LFP/GF-PPZ-16/Li cell shows stable and highly reversible charging/discharging processes based on only three cycles of CV testing (line 268-271). Conducting the CV test for more than ten cycles would be necessary to demonstrate the stability and reversibility of charging/discharging.

9 It is recommended to include TGA and DSC curves of the pristine separator (GF) and provide a more detailed discussion to understand the thermal behavior of the GF-PPZ-16 electrolyte when compared to the pristine separator. It is also suggested to present the thermal shrinkage of the prepared electrolyte.

10 Some corrections are needed. (a) The sample names (PPZ and GF-PPZ) should be identified consistently in parts such as non-flammability, ion conduction, t_{Li^+} , Li plating/stripping test discussion. (b) Table S5 should also include a comparison of some kinds of liquid electrolytes.

Response to Reviewers

Reviewer #1.

Reviewer #1's Comment: *This paper reports non-flammable liquid electrolytes for applications to Li-metal batteries. Overall, the characterization and practical application of liquid-state electrolytes are systematically investigated, and the results look promising. I think it should be published after minor revisions, as listed below.*

Author's reply: Thank you very much. We sincerely appreciate your recognition and suggestions. Following your instructive comments, we have revised our manuscript. Hopefully, we have addressed your concerns.

Reviewer #1's Comment 1: *It seems LPE exhibited a wide electrochemical window up to 5.2 V, potentially benefiting the application into the battery cells with high potential cathodes. The authors demonstrated the feasible implementation of LPE into battery cells with high-potential cathodes, but it shows significant capacity degradation over the cycle even though the loss in Coulombic efficiency and interfacial issue are negligible. I think the physic underlying such a high potential window is not thoroughly studied in this paper, likely helping to understand the capacity degradation. Further, even though the electrolytes have a high potential window, it does not indicate the stability in high potential batteries as there is an interfacial issue between the cathode and electrolyte. I think the use of porous electrodes for an electrochemical window test would help to mimic the actual situation.*

Author's reply: Many thanks to you for the precious comment and suggestion. We agree with your views on the difference between the electrochemical stability windows (ESWs) test value and practical application. The electrolyte stability is influenced by the chemistry and morphology of the working electrode (or cathode), and therefore the intrinsic stability of the electrolyte may not reflect the apparent stability in real cycling conditions¹.

Although testing the ESWs with the flat plate electrodes was different from the practical application of the high-potential cathode-based batteries, the original results were still retained to facilitate comparison with similar literature (They all used the same test method). At the same time, an extra LSV test based on **porous electrodes** was added in **Supplementary Figure 5** (*Supplementary Information*) for a multifaceted comparison.

To thoroughly study the physical basis of the high ESW of PPZ materials, theoretical calculations based on the first-principles density functional theory (DFT) were performed to analyze HOMO-LUMO orbitals of the simplified PPZ models. More experimental details and explanations are given below.

Q1: Electrochemical window test based on a porous electrode.

Modification:

The revised part of *the manuscript* is as follows:

(Lines 219-221) The LPEs exhibited outstanding electrochemical stabilities with wide electrochemical stability windows (ESWs) of up to 5.20 V based on flat plate electrodes

(or 4.55V based on graphite porous electrodes) at 90 °C, which exceeded those of LEs and PEO (Supplementary Fig.5).

The revised Supplementary Figure 5 in *Supplementary Information* is as follows:

(Page 5, *Supplementary Information*).....The graphite porous electrode was prepared by a reported method with minor modifications⁷. The polymer (PPZ or PEO) and graphite particles were uniformly dispersed in acetonitrile with a 1:1 mass ratio to form a 20 wt% concentration suspension, and then 50 μ L of the suspension was applied dropwise to a stainless-steel plate with a diameter of 15.5 mm and dried completely to form a graphite porous electrode.

(Page 14, *Supplementary Information*)

Supplementary Figure 5. Electrochemical windows tests at 90 °C. The LSV curves of

(a) PPZ-based LPEs, (b) PEO-based SPEs, and (c) commercial LE with stainless steel flat plate electrodes. (d) The LSV curves of GF-FPPZ-16 and GF-PEO-16 electrolytes with graphite porous electrodes. GF as a separator.

The PPZ-based LPEs exhibited a wide electrochemical window of 5.20 V based on flat plate electrodes at 90°C, which exceeded LE's 3.16 V and PEO-based SPE's 4.92 V. Notably, commercial LE exhibited poor oxidation stability, and the electrochemical window decreased significantly with increasing temperature in Supplementary Figure 5c. Compared with the flat electrode, the porous electrode is closer to the actual situation. As shown in Supplementary Figure 5d, the carbon porous electrode made the electrochemical window visibly lower by accelerating the interfacial reaction kinetics¹¹, which was due to the catalysis and larger interfacial contact^{7,12,13}. Despite this, GF-PPZ-16 still exhibited a electrochemical window of up to 4.55V, significantly better than GF-PEO-16's 3.30V.

Q2: Further research the physic underlying on the high potential window of PPZ.

Modification:

The revised part of *the manuscript* is as follows:

(Lines 224-240)Therefore, the LPEs are expected to be suitable for use in high-voltage lithium metal battery systems exhibiting higher energy storage densities. The oxidation potential in polymer electrolyte is mainly determined by the oxidation of polymer host⁴⁰, but performing computational works on entire polymer chains becomes prohibitive⁴¹. The fragment model can reveal the role of monomer chemistry in ESW⁴².

Therefore, the first-principles density functional theory (DFT) was used to calculate the redox potential and ESW width (the gap between oxidation potential and reduction potential) of simplified fragment single-chain model of polyphosphazene with different side chain lengths in Supplementary Fig.6. As the side chain became longer, the oxidation potential increased from 5.94 eV to 6.09 eV and the reduction potential decreased slightly, the ESW width increased from 7.22 eV to 7.47 eV. Therefore, the oxidation stability of PPZ can be considered to increase with the increase of side chain length. The illustration in Supplementary Fig.6 showed that the highest occupied molecular orbital (HOMO) levels were mainly distributed along the N/P main chain, while the lowest unoccupied molecular orbital (LUMO) levels were confined to the terminal. The shielding effect of long ether side chain with high oxidation stability on the lone electron on N atom may be the reason for the improvement of oxidation stability. In addition, it should be emphasized that the real ESW is affected by the electrode and lithium salt^{40, 43},.....

The revised part and newly added Supplementary Figure 6 in *Supplementary Information* are as follows:

(Page 7, Supplementary Information)

Density functional theory (DFT) calculations.

DFT calculations were performed on ORCA 5.0.3 program⁸ based on a simplified fragment chain model of PPZ². All structures were optimized in the B3LYP hybrid exchange-correlation functional as well as 6-31G* basis sets⁹. The vdW interactions were described using DFT-D3(BJ) dispersion correction. Moreover, The Visual

Molecular Dynamics (VMD) and Multiwfn software were used for analysis and mapping¹⁰.

(Page 15, Supplementary Information)

Supplementary Figure 6. Change in oxidation, reduction potentials and electrochemical stability windows with polyphosphazene side chain structure. DFT calculations were performed on fragment chain models on the right. Pictures show the position of the highest occupied molecular orbital (HOMO) and the lowest unoccupied molecular orbital (LUMO) levels. Color code: golden-carbon, white-hydrogen, red-oxygen, orange-phosphorus, blue-nitrogen.

Reviewer #1's Comment 2: *Even though it is a liquid electrolyte, the ionic conductivity at RT is very low. Please rationale it. This drives to very low capacity in the battery cells compared to a theoretical value at 30 and 40 degrees Celsius.*

Author's reply: Thanks to the reviewer for precious comment. It is well known that

polymers do not diffuse as freely as small molecules because of their large molecular structure, and the movement of polymer chains is temperature-dependent. PPZ was in viscous flow state at room temperature. Through the molecular thermal motion, PPZ realizes macroscopic flow like a liquid, only if the barycenter of PPZ has a relative displacement. However, the high viscosity of PPZ (**Figure 2c** of the manuscript) makes ion transport by molecular displacement still very limited. Therefore, the transport of ions in polymer electrolyte was achieved by chain segment motion and jumping between chains^{2, 3}. And the ion transport in polymer electrolytes can be described by the Vogel-Tamman-Fulcher (VTF) relationship⁴. As shown in **Figure 3f** of the manuscript, LPE satisfied VTF relationship rather than Arrhenius equation. In conclusion, the ion transport of LPE follows the rule of polymer electrolyte, and of course its ionic conductivity was inferior to that of LE due to its high viscosity and limited chain movement at room temperature. Nevertheless, the liquid properties of LPE can provide better interfacial contact and larger free volume, making the room temperature ionic conductivity of LPE reach the forefront of previously reported polymer electrolytes.

Reviewer #1's Comment 3: *Please calculate the charge passed through from the Galvanostatic cycling, which would help to compare the result with the literature value.*

Author's reply: We sincerely acknowledge your kind review and valuable suggestions.

We re-labeled the current density and charge quantity in the graphic and text description of the galvanostatic polarization cycling in **Figure 3g** and **h** of the manuscript.

Modification:

The revised part of the manuscript is as follows:

(Lines 199-210)

Fig. 3. Chemical and electrochemical properties of LPE electrolytes. a, FTIR spectra of LPEs. **b,** Raman characteristic peaks of TFSI in LPEs after fitting (right), the curve was normalized by the signal of CH₂ at 1455.7 cm⁻¹ (left). **c,** DSC curves of LPEs. **d,** Radial distribution functions g(r) of Li-O and Li-N pairs from MD simulation of PPZ-16. **e,** Coordination structure of Li⁺ and PPZ in PPZ-16. **f,** Ionic conductivity of GF-LPEs with different Li-salt concentrations as a function of temperature, the dashed lines represent the VTF fitting results. **g,h,** Voltage profiles of lithium-metal plating/stripping in Li//Li symmetrical cells with (g) constant current density of 0.1 mA

cm⁻² (capacity: 0.1 mAh cm⁻²) and (h) variable current density from 0.05 to 0.5 mA cm⁻² (capacity: 0.2 mAh cm⁻²) at 90 °C. The insets are voltage profiles of Li/LPE/Li cells at 5, 105, 162 and 2005h, respectively.

Reviewer #1's Comment 4: *Please explain the reason for weak interfacial polarization in LPE-electrode interface (charge and discharge curve).*

Author's reply: Thank you very much for your precious comment. We have elaborated this part in more detail in the manuscript.

Modification:

The revised part of *the manuscript* is as follows:

(Lines 364-371)

.....The corresponding specific capacity-voltage curves also showed typical charge–discharge profiles between 2.5 and 4.2 V (Supplementary Fig.22). The overpotential was about 93mV at 0.2C, reflecting good interface contact and fast kinetics of Li⁺ transport^{48,49}, corresponding to low ohmic polarization and activation polarization. As the current density increased, the concentration polarization and activation polarization became apparent resulting in the overpotential increasing to 377mV at 2C, but still within the acceptable range. The low interfacial polarization was attributed to the good wettability of the liquid state LPE to the electrodes, as well as the stable electrode/electrolyte interface derived from LPE.....

Reviewer #1's Comment 5: *The battery cells showed very stable cyclability at 120*

degree Celsius, but the significant capacity degradation was observed at 60 and 90 degree Celsius even though the loss in Coulombic efficiency and interfacial issue are negligible. Please rationale it.

Author's reply: Thank you very much for your reminding. Actually, the cycle number and current density for the cyclic performance diagram at different temperatures in **Figure 4h-j** of the manuscript are different. By comparison (see below **Figure 1**), the capacity retention after 100 cycles at 60°C, 90°C and 120°C were 98.80%, 97.17% and 95.87%, respectively, indicating the negligible difference of their capacity degradation.

Figure 1. The performance of LFP/GF-PPZ-16/Li cells for 100 cycles at different temperatures.

Reviewer #1's Comment 6: *Please provide the adhesion energy of GP for comparison.*

Author's reply: Thank you for your precious suggestion. We tested the adhesion energy of electrolyte to graphite (GP or Gr) anode. A histogram with error bars (**Supplementary Figure 27a**) was provided to make the test data more precise.

Modification:

The revised part of *the manuscript* is as follows:

(Lines 407-409)Therefore, an interfacial adhesion test was used to quantitatively describe the strength of the interfacial contact⁴⁷. The adhesion energies of GP-PPZ-16 sandwiched between the LFP cathode and Cu anode or graphite anode reached 183.8 ± 24.42 and 106.03 ± 33.63 J m⁻² (Supplementary Fig. 27), respectively.....

The revised part in *Supplementary Information* is as follows:

(Pages 4 and 5, Supplementary Information)

.....Cathode and anode were cut into 20×80mm rectangles and GF-PPZ-16 was cut into 20×15mm rectangles. Then align the three along one end and fit them together. The complete sandwich structure was placed at 90°C for 2 hours to fully bond the battery structure.....

Supplementary Figure 27. Adhesion test of GF-PPZ-16 LPE to two battery structure. **a**, Histogram with error bar for adhesion energy. **b**, The optical image of the test. Adhesion test data of (c) LFP/GF-PPZ-16/Cu cell and (d) LFP/GF-PPZ-16/Gr cell, the inserts show the electrodes after adhesion test. The adhesion of LPE to electrodes was higher than the structural strength of the self-made graphite anode, resulting in the graphite electrode being stripped from the copper foil during the test.

Reviewer #2.

Reviewer #2's Comment: *The manuscript entitled "Non-flammable solvent-free liquid polymer electrolyte for lithium metal batteries" reports on stable long-term cycling of lithium metal batteries at high temperatures (60 and 90°C) using a solvent-free liquid*

poly[bis-(methoxyltriethoxy) phosphazene] (PPZ) with non-flammability and high thermal stability. However, the current version of the manuscript contains only limited data and lacks a comprehensive discussion. Therefore, the authors need to provide more data and explanations to optimize the amount of salt and improve the electrochemical performance. In addition, the organization of data is inconsistent among conductivity, electrochemical stability, and cell performance. Based on these issues, it is not recommended for publication in the current form. The following issues need to be addressed:

Author's reply: Thank you very much for the positive and precious comments on our paper. We understand your careful consideration and have tried our best to address your proposal. We sincerely appreciate your suggestions. These suggestions have played a crucial role in the improvement of this manuscript, especially in some details that we have not noticed. We carefully analyzed your comments and made extensive additions and revisions to the manuscript. Hopefully, we have addressed your concerns and this manuscript could reach the level of publication.

Reviewer #2's Comment 1: *The filling of PPZ into GF separator needs to be demonstrated more clearly. This includes more experiments for filling PPZ into GF separator; explanation of how to fill GF separator with PPZ given its viscosity, what kind of solvent was used for this preparation, and evidence such as SEM-EDS mapping and contact angle between PPZ and GF separator in Fig. S8. Porosity before and after filling PPZ into GF separator should be investigated, and the filling percentage (or*

PPZ uptake) should be determined. The authors should also show data to prove whether there is any leakage of PPZ after filling.

Author's reply: We sincerely acknowledge your precious comments and valuable suggestions. In order to make the whole preparation process of LPEs clear, we provided more details. It should be acknowledged that although LPEs were fluid at room temperature, they were too viscous to be used directly like LEs. In order to ensure the efficiency and quality of the filling, ethylene glycol dimethyl ether (DME) was used as an auxiliary solvent and subsequently was completely removed by heating under vacuum. **Supplementary Figure 15** of Supplementary Information provided a more detailed preparation flow chart. Preparation details and LPE load were added in the preparation section of Supplementary Information. In addition, we provided the SEM-EDS mapping of the GF-PPZ-16 in **Supplementary Figure 16** of Supplementary Information, and an additional heating experiment in **Supplementary Figure 19** of Supplementary Information to prove that no leakage occurred. Contact angle testing was not available because the highly viscous LPEs cannot be extruded from the syringe to form spherical droplets. Therefore, a infiltration experiment was provided in **Supplementary Figure 14** of Supplementary Information.

Modification:

The revised part of the manuscript are as follows:

(Lines 316-330) To assess the feasibility of using the LPEs in practical batteries, the LPEs were loaded on GF separators to assemble Li metal full batteries containing a LiFePO₄ (LFP) cathode (Fig.4a). **Supplementary Fig.14 showed that under heating at**

80°C, LPE (PPZ-40) can infiltrate gradually to GF separator for 20 min and spread out fully in GF after 5 hours. Therefore, the relatively high viscosity made the direct use of LPE for battery assembly inefficient. In order to ensure the uniformity and consistency of electrolytes in separators, LPEs were loaded on separators by a solvent-assisted method (Supplementary Fig.15). The solvent (ethylene glycol dimethyl ether) with a low boiling point can be completely removed by vacuum heating. The GF separator filled with the LPE, denoted as GF-LPE, showed good flexibility and completely returned to its original state after 180° folding (Fig.4b). Specifically, SEM and EDS images showed that PPZ-16 was uniformly filled into GF separator to form a dense and smooth structure (Fig.4c and Supplementary Fig.16). Similar results were found for cellulose membrane (CM) and polypropylene (PP) separators (Supplementary Fig.17, S18), which indicated that the LPEs were suitable for various commercial separators. In addition, a heating experiment demonstrates that GF-LPE has good dimensional stability and there is no leakage of LPE at 25-180°C (Supplementary Fig.19).

The revised parts and Supplementary Figure 14, Figure 15, Figure 16, Figure 19 in *Supplementary Information* are as follows:

(Page 3, Supplementary Information)

.....For practical use, LPEs were filled into separators (such as GF, CM, or PP membranes). To ensure filling uniformity and efficiency, 75 wt% DME solvent was used to further reduce the viscosity of LPEs to achieve adequate fluidity. Then, the mixed solution (600 µL) was dropped into the GF separator (φ19 mm). Similarly, the mixed solution (100 µL) was added to the PP separator or CM separator. After the

solvent evaporated in the air, LPEs-loaded separators were heated to 100°C for 12h to make LPEs evenly filled. Then, all LPEs-loaded separators were vacuum dried at 80°C for more than 48 hours to ensure that solvent was removed fully. The load of LPEs in GF, CM, and PP were (121.00±0.22) mg, (20.78±0.04) mg, and (19.25±0.06) mg, respectively.....

(Page 23, Supplementary Information)

Supplementary Figure 14. Infiltration experiment of LPE (PPZ-40) to GF at 80°C. No solvent was involved in this process.

(Page 24, Supplementary Information)

Supplementary Figure 15. Schematic diagram of loading LPEs onto a separator by the solution-assisted method.

Supplementary Figure 16. EDS images on surface and section of GF-PPZ-16 electrolyte.

(Page 28, Supplementary Information)

Supplementary Figure 19. Vertical heating test of GF-LPEs. Used to determine electrolyte leakage and thermal deformation. **a**, Schematic diagram of vertical heating test, GF-LPE was adhered to the glass slide by its own adhesion. **b**, Photos of the test process. **c**, Samples photos after heating different times at various temperatures.

Reviewer #2's Comment 2: *In LSV investigation, the authors need to provide more detailed explanations of why the GF-PPZ-16 (O:Li⁺ ratio of 16:1) has much higher electrochemical stability (5.2 V) when compared to the GF-LE (4.2 V). The full-profile (containing lower potential limit) of LSV curves should be provided to evaluate the Li plating/stripping behavior.*

Author's reply: Thank you very much for your precious comments. According to your suggestion, we modified the relevant graphs. The full-profile (containing lower potential limit) of LSV curves were provided. LSV curves of GF-LE at 25°C, 60°C, and 90°C were provided in **Supplementary Figure 5c** of Supplementary Information. It

can be seen that the oxidation stability of GF-LE decreased significantly with increasing temperature, indicating that the electrochemical stability of LE is highly sensitive to temperatures. In addition, through theoretical calculation, we further discussed the essential reasons for the high electrochemical stability of PPZ.

Modification:

The revised parts of the manuscript are as follows:

(Lines 224-240)

.....Therefore, the LPEs are expected to be suitable for use in high-voltage lithium metal battery systems exhibiting higher energy storage densities. The oxidation potential in polymer electrolyte is mainly determined by the oxidation of polymer host⁴⁰, but performing computational works on entire polymer chains becomes prohibitive⁴¹. The fragment model can reveal the role of monomer chemistry in ESW⁴². Therefore, the first-principles density functional theory (DFT) was used to calculate the redox potential and ESW width (the gap between oxidation potential and reduction potential) of simplified fragment single-chain model of polyphosphazene with different side chain lengths in Supplementary Fig.6. As the side chain became longer, the oxidation potential increased from 5.94 eV to 6.09 eV and the reduction potential decreased slightly, the ESW width increased from 7.22 eV to 7.47 eV. Therefore, the oxidation stability of PPZ can be considered to increase with the increase of side chain length. The illustration in Supplementary Fig.6 showed that the highest occupied molecular orbital (HOMO) levels were mainly distributed along the N/P main chain, while the lowest unoccupied molecular orbital (LUMO) levels were confined to the terminal. The

shielding effect of long ether side chain with high oxidation stability on the lone electron on N atom may be the reason for the improvement of oxidation stability. In addition, it should be emphasized that the real ESW is affected by the electrode and lithium salt^{40, 43}.

The revised Supplementary Figure 4 and Figure 5 in *Supplementary Information* are as follows:

(Page 14, Supplementary Information)

Supplementary Figure 5. Electrochemical windows tests at 90 °C. The LSV curves of (a) PPZ-based LPEs, (b) PEO-based SPEs, and (c) commercial LE with stainless steel flat plate electrodes. (d) The LSV curves of GF-FPPZ-16 and GF-PEO-16 electrolytes with graphite porous electrodes. GF as a separator.

The PPZ-based LPEs exhibited a wide electrochemical window of 5.20 V based on flat plate electrodes at 90°C, which exceeded LE's 3.16 V and PEO-based SPE's 4.92 V. Notably, commercial LE exhibited poor oxidation stability, and the electrochemical window decreased significantly with increasing temperature in Supplementary Figure 5c. Compared with the flat electrode, the porous electrode is closer to the actual situation. As shown in Supplementary Figure 5d, the carbon porous electrode made the electrochemical window visibly lower by accelerating the interfacial reaction kinetics¹¹, which was due to the catalysis and larger interfacial contact^{7,12,13}. Despite this, GF-PPZ-16 still exhibited a electrochemical window of up to 4.55V, significantly better than GF-PEO-16's 3.30V.

(Page 7, Supplementary Information)

Density functional theory (DFT) calculations.

DFT calculations were performed on ORCA 5.0.3 program⁸ based on a simplified fragment chain model of PPZ². All structures were optimized in the B3LYP hybrid exchange-correlation functional as well as 6-31G* basis sets⁹. The vdW interactions were described using DFT-D3(BJ) dispersion correction. Moreover, The Visual Molecular Dynamics (VMD) and Multiwfn software were used for analysis and mapping¹⁰.

(Page 15, Supplementary Information)

Supplementary Figure 6. Change in oxidation, reduction potentials and electrochemical stability windows with polyphosphazene side chain structure. DFT calculations were performed on fragment chain models on the right. Pictures show the position of the highest occupied molecular orbital (HOMO) and the lowest unoccupied molecular orbital (LUMO) levels. Color code: golden-carbon, white-hydrogen, red-oxygen, orange-phosphorus, blue-nitrogen.

Reviewer #2's Comment 3: *The authors need to discuss the mechanism of Li-ion transport more clearly, including the effect of O:Li⁺ ratio on ionic conductivity in more detail at RT, 60, 90, and 100°C. The increasing trend of ionic conductivity was not consistent at RT, 60, and 90°C. The highest ionic conductivity at RT, 60, and 90°C was obtained from GF-PPZ-40, GF-PPZ-24, and GF-PPZ-16, respectively (see Table 4). The authors need to explain the reason for this difference in the highest ionic conductivity vs. temperature at various O:Li⁺ ratios.*

Author's reply: Thank you very much for your precious comments. We are very sorry that we omitted a detailed discussion of this part in the original manuscript. This part was complicated by the coupling of lithium concentration and temperature on ionic conductivity. According to your suggestions, we tried our best to elaborate on this part in more detail, hoping to solve your doubts.

Modification:

The revised parts of the manuscript are as follows:

(Lines 249-267)

.....As expected, the LPEs exhibited excellent ion transport abilities, especially at room temperature, which were 1-2 orders of magnitude greater than that of PEO^{21,43}.

Generally, the σ of electrolyte is the product of ionic mobility (μ) and free carrier concentration (c)¹⁹. Ion transport through polymer electrolytes is intimately coupled to the motion of the polymer chain. The activation energy (E_a) was calculated with the Vogel–Tamman–Fulcher (VTF) empirical equation. As shown in Fig.3f and Table 4, the E_a values for GF-PPZ-8, GF-PPZ-16, GF-PPZ-24, and GF-PPZ-40 were 26.11, 11.36, 8.56, and 8.06 kJ mol⁻¹, respectively. Good linear fitting relationship showed that ion transport mechanism in LPEs was consistent with that in SPEs. The E_a value reflected the influence of temperature on the ion transport. Lithium salt concentration affected the movement of polymer chain segments to varying degrees, resulting in the difference of activation energy. Since both lithium salt content and temperature have effects on carrier concentration and ion mobility, although it becomes difficult to discuss single variables, the changing trend of ionic conductivity with temperature at different O:Li⁺

ratios was qualitatively analyzed. At room temperature, GF-PPZ-40 has the highest σ due to its lowest E_a . With the increase of temperature, the movement ability of polymer chain enhanced, and the σ was controlled by both carrier concentration and activation energy. Therefore, at 60°C, the σ of GF-PPZ-24 and GF-PPZ-16 were greatly increased, but the increase of GF-PPZ-40 was limited. At 90°C, the influence of activation energy became more weaker, while the carrier concentration was dominated, resulting in GF-PPZ-16 showing the largest σ

Reviewer #2's Comment 4: *In Li-ion transference number (t_{Li^+}) analysis, the authors need to explain why t_{Li^+} increased from 0.1 to 0.3 with increasing O:Li⁺ ratios from 16 to 40. The t_{Li^+} of GF-LE should be shown and compared with GF-PPZ electrolytes. Additionally, the authors should explain why they conducted t_{Li^+} tests at 90°C while other properties (LSV, cell performance) were measured at RT and 60°C. The t_{Li^+} of GF-PPZ and GF-LE should be provided at RT and 60°C.*

Author's reply: Thank you very much for your precious comments and valuable suggestions. In the revised manuscript, we have discussed in more detail the change of t_{Li^+} with lithium salt content, and sincerely hope to solve your doubts. According to your suggestion, we have tested the t_{Li^+} of GF-LE. At 25°C, the t_{Li^+} of GF-LE was 0.74. However, when the temperature rose to 60°C, we fail to obtain an effective migration number of GF-LE due to the abnormal EIS data. This anomaly could be attributed to the significant electrochemical degradation of LE at a high temperature. Considering that GF-LPE and GF-LE have completely different ion transport mechanisms, in our

opinion, comparing the t_{Li^+} of GF-LPE with that of GF-LE is not necessary in manuscript. In addition, we are very sorry to you for the trouble caused by the measured temperature. We did our best to avoid these problems in the revised manuscript. Actually, all electrochemical performance (LSV, σ , t_{Li^+} , Galvanostatic plating/stripping test, CV, cell performance, etc.) were provided at 90°C, which was the main temperature of this work to discuss the thermal stability and safety of LPEs. However, the t_{Li^+} of GF-PPZ from 25 °C to 90 °C were provided in the revised manuscript and **Supplementary Table 5**. We sincerely hope that we have addressed your concerns. The modified parts in the revised manuscript are as follows:

Modification:

The revised parts of the manuscript are as follows:

(Lines 269-282)

.....Specifically, at 90°C, GF-PPZ-8 reached the largest t_{Li^+} of 0.38, and with a further increase in the O:Li⁺ ratio from 16 to 40, as with PPZ-16 and PPZ-40, the t_{Li^+} decreased from 0.30 to 0.1 (Supplementary Fig.8). The t_{Li^+} of various GF-PPZ at a range of RT (25 °C) to 90 °C were tested and shown in Supplementary Table 5, indicating no significant temperature dependence. The low t_{Li^+} can be attributed to the “pocket effect”^{45,46}. As described in Supplementary Fig.9 (Molecular dynamics simulation), Li⁺ was bound into PPZ by Li-O and Li-N bonds, while TFSI⁻ anions were distributed around the polymer chain due to lack of interaction and large steric hindrance. Therefore, under an electric field, anions have a faster migration capacity resulting in a low t_{Li^+} . Especially in PPZ-40, LiTFSI was completely dissociated by

PPZ (Fig. 3b and Supplementary Table 1), meaning that all free Li^+ was bonding, resulting in the lowest t_{Li^+} . With the increase of LiTFSI content, the non-free state Li^+ (CIP and AGG) was considered to weaken the interaction between Li^+ and PPZ and thus increased the transference number.....

The revised Supplementary Table 5 in *Supplementary Information* is as follows:

(Page 46, *Supplementary Information*)

Supplementary Table 5. The lithium-ion transference number (t_{Li^+}) and effective Li^+ conductivity (σ_{Li^+}) of LPEs with different LiTFSI content at various temperatures. (σ_{Li^+} :

S cm^{-1})

Temp. °C	GF-PPZ-8		GF-PPZ-16		GF-PPZ-24		GF-PPZ-40	
	t_{Li^+}	σ_{Li^+}	t_{Li^+}	σ_{Li^+}	t_{Li^+}	σ_{Li^+}	t_{Li^+}	σ_{Li^+}
25	-	-	0.37	1.33E-05	0.16	1.24E-05	0.11	1.20E-05
30	-	-	0.27	1.21E-05	0.20	1.78E-05	0.09	1.03E-05
40	-	-	0.28	2.78E-05	0.16	2.67E-05	0.09	1.66E-05
50	-	-	0.30	5.40E-05	0.17	4.10E-05	0.09	2.43E-05
60	0.46	4.00E-06	0.31	9.67E-05	0.16	6.61E-05	0.10	3.58E-05
70	0.44	9.55E-06	0.34	1.75E-04	0.16	8.80E-05	0.12	6.94E-05
80	0.39	1.71E-05	0.32	2.78E-04	0.15	1.01E-04	0.10	7.47E-05
90	0.38	3.15E-05	0.30	4.02E-04	0.16	1.57E-04	0.10	8.64E-05

The revised Supplementary Figure 9 in *Supplementary Information* are as follows:

(Page 18, Supplementary Information)

Supplementary Figure 9. Molecular dynamics (MD) simulation of the distribution of Li⁺ and TFSI⁻ in LPE.

Reviewer #2's Comment 5: *The authors need to show all data of the effective Li-ion conductivity at various temperatures fully along with more discussion to prove that the GF-PPZ-16 has the highest effective Li-ion conductivity.*

Author's reply: Thank you very much for your precious suggestions. To study the performance of LPE-based batteries in high temperature environment, we mainly focused on the electrochemical measurements of LPE at 90 °C. But to address your concerns, we try our best to add these data at all temperatures, which were shown in

Supplementary Table 5 of Supplementary Information. Notably, GF-PPZ-16 showed the highest effective ionic conductivity at most temperatures, corresponding to the best comprehensive electrochemical performance of cells with GF-PPZ-16.

Modification:

The revised part of **the manuscript** was as follows:

(Lines 282-287).....The effective Li^+ conductivity (σ_{Li^+}) was calculated in **Supplementary Table 5** at various temperatures. Due to its good performance in both σ and t_{Li^+} , GF-PPZ-16 had the highest σ_{Li^+} at nearly all temperatures. At 90°C , the σ_{Li^+} of GF-PPZ-16 reached $4.02 \times 10^{-4} \text{ S cm}^{-1}$, and the σ_{Li^+} of GF-PPZ-8, GF-PPZ-24 and GF-PPZ-40 were 3.15×10^{-5} , 1.57×10^{-4} , $8.64 \times 10^{-5} \text{ S cm}^{-1}$, respectively.....

The revised **Supplementary Table 5** in *Supplementary Information* is as follows:

(Page 46, *Supplementary Information*)

Supplementary Table 5. The lithium-ion transference number (t_{Li^+}) and effective Li^+ conductivity (σ_{Li^+}) of LPEs with different LiTFSI content at various temperatures. (σ_{Li^+} : S cm^{-1})

Temp. °C	GF-PPZ-8		GF-PPZ-16		GF-PPZ-24		GF-PPZ-40	
	t_{Li^+}	σ_{Li^+}	t_{Li^+}	σ_{Li^+}	t_{Li^+}	σ_{Li^+}	t_{Li^+}	σ_{Li^+}
25	-	-	0.37	1.33E-05	0.16	1.24E-05	0.11	1.20E-05
30	-	-	0.27	1.21E-05	0.20	1.78E-05	0.09	1.03E-05
40	-	-	0.28	2.78E-05	0.16	2.67E-05	0.09	1.66E-05
50	-	-	0.30	5.40E-05	0.17	4.10E-05	0.09	2.43E-05
60	0.46	4.00E-06	0.31	9.67E-05	0.16	6.61E-05	0.10	3.58E-05
70	0.44	9.55E-06	0.34	1.75E-04	0.16	8.80E-05	0.12	6.94E-05
80	0.39	1.71E-05	0.32	2.78E-04	0.15	1.01E-04	0.10	7.47E-05
90	0.38	3.15E-05	0.30	4.02E-04	0.16	1.57E-04	0.10	8.64E-05

Reviewer #2's Comment 6: *The authors need to clarify why Li-dendrite growth was suppressed in the Li/PPZ-16/Li cell. Some evidence should be provided, such as SEM and XPS analysis of Li metal before and after 2500 h cycling.*

Author's reply: Thank you very much for your precious suggestion. This part was omitted because we had systematically discussed the interface problems of LFP/GF-PPZ-16/Li cell in the section of *Morphology and chemistry at the interphase* of the manuscript. We believe that your proposal is necessary. Therefore, in accordance with your suggestion, we have added more experimental evidences (SEM, EDS, and XPS) in **Supplementary Figure 10-12** of Supplementary Information, and further clarified it in the corresponding part of the manuscript.

Modification:

The revised part of the manuscript is as follows:

(Lines 289-302) Interestingly, the LPEs also exhibited excellent inhibition of lithium dendrite formation. The galvanostatic cycling tests demonstrated low Li plating/stripping overpotentials at 90°C in Li/LPEs/Li symmetrical cells. These cells exhibited ultralong stable cycles over 2200 h, which were only accompanied by very slight fluctuations in voltage (Fig. 3g). The magnification of voltage curves at different times showed that there was no micro-short circuits or voltage drops caused by lithium dendrites⁴⁷. Scanning electron microscopy (SEM) images in Supplementary Fig. 10 showed a dense interface layer on the surface of the lithium metal electrode after cycling, and the cross-section images showed that the deposition of Li⁺ mainly occurred below the interface layer. Energy dispersive spectrometer (EDS) images showed that

the elements were evenly distributed in the interface (Supplementary Fig. 11). X-ray photoelectron spectroscopy (XPS) revealed that the interface was composed of organic fragments and inorganic lithium compounds derived from PPZ and LiTFSI (Supplementary Fig. 12), mainly containing LiF, Li₃N, Li₂CO₃, Li₃PO₄, etc. This dense hybrid interface ensured good transport of Li⁺ and impeded the growth of lithium dendrites.

The revised parts in *Supplementary Information* are as follows:

(Page 19, *Supplementary Information*)

Supplementary Figure 10. SEM images of (a,b) surface and (c,d) cross-section of Li electrode in Li/GF-PPZ-16/Li cell after 2200h operation.

(Page 11, Supplementary Information)

Supplementary Figure 11. SEM-EDS images of (a) cross-section and (b) surface of Li electrode in Li/GF-PPZ-16/Li cell after 2200h operation.

(Page 21, Supplementary Information)

Supplementary Figure 12. XPS spectra of Li electrode surface in Li/GF-PPZ-16/Li

cell after 2200h operation.

Reviewer #2's Comment 7: *Why is GF-PPZ-16 light yellow in color in Fig. S8 while others are white? The scale bar in the cross-sectional SEM images is not consistent between Fig. S8 and Fig. S9. It is not convincing to claim that the GF separator is the best choice among GF, CM, and PP separators solely based on SEM images showing its dense and smooth structure. More information on the effective Li-ion conductivity of the separators fully dipped in PPZ-16 should be provided for a fair comparison.*

Author's reply: Thank you very much for your precious comments and suggestions.

We feel sincerely sorry for the inconvenience brought to the reviewers. The color difference of the samples was due to the light when taking pictures. The original LPEs were dark yellow. Influenced by the color of the separator and background, the exact color of GF-LPEs was between white and buff as shown in **Supplementary Figure 17** of Supplementary Information. In order to avoid misunderstanding, we have replaced the relevant pictures.

Since GF is several times thicker than PP and CM, the scale bars in **Supplementary Figure 8 and 9** do not fully show the cross-sectional morphology of GF-PPZ-16 and GF. Therefore, the inserts having a scale bar being consistent with CM-PPZ-16 and PP-PPZ-16 as well as PP and CM were given in the cross-sectional images of GF-PPZ-16 and GF.

Because the GF is porous, nonflammable, and chemically stable, it will not affect the conclusion of the study on LPEs, especially in high-temperature stability and safety. In fact, GF has the highest n-butanol absorption capacity (190.05% vol) relative to PP (71.35% vol) and CM (36.63% vol). PP will shrink at high temperatures, and too small vertical pores in PP are very unfavourable for the filling of LPE. The abundant hydroxyl groups of CM are detrimental to the cycle stability of batteries. Overall, in consideration of the minimal impact on LPEs' properties, GF was selected for the study of this work.

Modification:

The revised parts in *Supplementary Information* are as follows:

(Page 26, Supplementary Information)

Supplementary Figure 17. a-c, The digital photographs and SEM images of the LPEs

with different separators: (a) GF-PPZ-16, (b) CM-PPZ-16, and (c) PP-PPZ-16.

(Page 27, Supplementary Information)

Supplementary Figure 18. **a-c**, The digital photographs and SEM images of different separators: (a) GF, (b) CM, and (c) PP.

Reviewer #2's Comment 8: *It is not possible to conclude that the LFP/GF-PPZ-16/Li cell shows stable and highly reversible charging/discharging processes based on only three cycles of CV testing (line 268-271). Conducting the CV test for more than ten cycles would be necessary to demonstrate the stability and reversibility of charging/discharging.*

Author's reply: Thank you very much for your precious suggestions. We updated a 10 cycles CV test to replace the original data in **Figure 4e** of the manuscript. With more cycles, the redox peaks appeared a certain degree of shift, which may be related to the interface construction and polarization. However, as the cycle progresses, the peak potential offsets gradually decreased, indicating that a stable interface was gradually

formed.

Modification:

The revised parts of the manuscript are as follows:

(Lines 331)

Fig. 4. Fabrication and performance of LPE-based cells.....e, CV curves of

LFP/GF-PPZ-16/Li cell with a scan rate of 0.2 mV s^{-1}

(Lines 356-359)

.....The oxidation and reduction peak potentials of all cells were approximately 3.75

and 3.08 V, respectively, which corresponded to delithiation and lithiation of the cathode. With more cycles, the peak potentials appeared a certain shift, which may be related to the interface construction and polarization. However, the peak potential offset decreased gradually, indicating a stable interface has been formed.....

Reviewer #2's Comment 9: *It is recommended to include TGA and DSC curves of the pristine separator (GF) and provide a more detailed discussion to understand the thermal behavior of the GF-PPZ-16 electrolyte when compared to the pristine separator. It is also suggested to present the thermal shrinkage of the prepared electrolyte.*

Author's reply: Thank you very much for your precious suggestions. The DSC and TGA curves of GF were provided in **Supplementary Figure 3, Supplementary Figure 4, Supplementary Table 2, and Supplementary Table 3** of Supplementary Information. At the same time, the detailed discussion was given in the revised manuscript. As shown in **Supplementary Figure 3**, no any thermal transition can be found in DSC curve of the GF, while both PPZ-16 and GF-PPZ-16 presented a glass transition with a low T_g , i.e. -39.66°C and -38.34°C , respectively. The T_g difference value between PPZ-16 and GF-PPZ-16 is only 1.32°C . Therefore, it is reasonable to suppose that the GF separator has no obvious effect on the thermal properties of PPZ-16. TGA curve in **Supplementary Figure 4** of Supplementary Information showed that the GF separator was thermally stable at $40\text{-}700^\circ\text{C}$ (the mass change was only 2.54wt%). Similarly, GF did not change the initial decomposition temperature ($T_{5\%}$) of PPZ-16. All in all, using GF did not change the thermal behavior of electrolytes.

In addition, additional thermal shrinkage from 25°C to 180°C was provided in **Supplementary Figure 19** of Supplementary Information.

Modification:

The revised parts of the manuscript are as follows:

(Lines 183-198)

.....Although the lithium salt increased the T_g of the LPE by occupying the free volume and coordinating, the T_g were still much lower than room temperature. The T_g of PPZ-8, PPZ-16, PPZ-24 and PPZ-40 were -31.49°C, -39.66°C, -53.60°C and -62.65°C, respectively. As shown in Supplementary Fig.3, the T_g of PPZ-16 containing GF separator (GF-PPZ-16) was -38.34°C, and no any thermal transition can be found in DSC curve of the GF, which means that introducing inert GF only caused a trivial change of 1.32 °C.

In addition, the thermal stability of the material determines the upper limit for the application temperatures. As shown in Supplementary Fig.4 and Table 3, PPZ presented outstanding thermal stability with an initial decomposition temperature (5% weight loss temperature, $T_{5\%}$) of 280°C. The $T_{5\%}$ of the PPZ-16 and GF-PPZ-16 were 230.0°C and 230.5°C, respectively, indicating that GF had no effect on the initial decomposition temperature of the electrolyte. The reason should be contribute to highly thermal stability of GF at 40-700°C, with a mass loss of only 2.54%. Compared with PPZ, for PPZ-16 and GF-PPZ-16, the decrease of $T_{5\%}$ might be due to the catalytic decomposition of the side groups by LiTFSI. Even so, $T_{5\%}$ of LPE was still much higher than the operating temperatures of LMBs.....

The revised parts in *Supplementary Information* are as follows:

(Page 12, *Supplementary Information*)

Supplementary Figure 3. DSC curves.

(Page 13, *Supplementary Information*)

Supplementary Figure 4. TG and DTG profile of LPEs.

(Page 43, Supplementary Information)

Supplementary Table 2. (continued)

Sample	GF	PPZ-16	GF-PPZ-16
$T_g/^\circ\text{C}$	-	-39.66	-38.34

* The number represents the molar ratio of the O atom and Li^+ ion in LPEs.

(Page 44, Supplementary Information)

Supplementary Table 3. The decomposition parameter of LPEs in nitrogen at a heating rate of $10^\circ\text{C min}^{-1}$.

Sample	$T_{5\%}(^\circ\text{C})$	$T_{max}(^\circ\text{C})$	MDR* (% min^{-1})	Residue at $700^\circ\text{C}(\text{wt}\%)$
GF	-	-	-	97.46
LiTFSI	374.1	448.0	-12.65	2.60
PPZ	280.0	297.9/496.6	-30.74/-1.53	12.8
PPZ-16	230.0	236.2/376.0	-17.3/-4.87	12.21
GF-PPZ-16	230.5	243.6/395.1	-15.85/-4.08	22.36

*MDR=Maximum decomposition rate.

Reviewer #2's Comment 10: *Some corrections are needed. (a) The sample names (PPZ and GF-PPZ) should be identified consistently in parts such as non-flammability, ion conduction, t_{Li^+} , Li plating/stripping test discussion. (b) Table 5 should also include a comparison of some kinds of liquid electrolytes.*

Author's reply: Thank you very much for your comments and precious suggestions.

(a) According to your suggestion, we have carefully revised the sample name to ensure consistency. (b) Some advanced liquid electrolytes were introduced for comparison in the **Supplementary Table 6** of Supplementary Information.

Modification:

The revised parts in *Supplementary Information* are as follows:

(Page 47, *Supplementary Information*)

Supplementary Table 6. The comprehensive performance of LPEs and recently reported advanced SPEs for LMBs.

Type	Electrolyte	σ / mS cm ⁻¹ (RT)	ESW/ V	Cell performance ^①	Operating temp./°C	Flame retardancy ^②
LE	EGDBE-E ⁵	4.01	N/A	~200mAh g ⁻¹ (NCM811), 200cycles at 0.5C, CLR≈0.1% ^③	60	0
	Pyrolux™-LiBOB/EC/PC/VC ⁶	3.38	5.0	155mAh g ⁻¹ , 35cycles at 0.33C, CLR<0.29%	120	1
	ADFN ⁷	N/A	4.65	~160mAh g ⁻¹ , 100cycles at 1C	90	0
SPE	m-PPL ⁸	0.0185	5.2	148.9mAh g ⁻¹ , 1000cycles at 1C, CLR=0.024%	60	0
	M-S-PEGDA ⁹	0.226	5.4	143.7mAh g ⁻¹ , 500cycles at 0.5C, CLR=0.029%	40	0
	PPC-TEGDME ¹⁰	0.89(60°C)	4.89	103mAh g ⁻¹ , 500cycles at 1C, CLR=0.046%	60	0
	PEO-PI-DBDPE ¹¹	0.0067	4.2	143mAh g ⁻¹ , 300cycles at 0.5C, CLR=0.03%	60	1
	IPN9-10PPC ¹²	0.033 (40°C)	4.2	141.5 mAh g ⁻¹ , 200cycles at 0.2C, CLR=0.037%	90	0
	Ultrathin-PI-PEO ¹³	0.23(30°C)	4.2	138 mAh g ⁻¹ , 200cycles at 0.5C.	60	0
LPE	This work	0.109	5.2	160.4mAh g ⁻¹ , 1000cycles at 0.5C, CLR=0.036%	60	1
				159mAh g ⁻¹ , 1000cycles at 1C, CLR=0.038%	90	1
				160mAh g ⁻¹ , 100cycles at 2C, CLR=0.066%	120	1

① Unmarked cathode defaults to LFP, and unmarked anode defaults to Li anode.

② 0 represents flammable, and 1 represents nonflammable.

③ CLR=Capacity loss rate (per cycle).

References.

1. Cabañero Martínez MA, *et al.* Are Polymer-Based Electrolytes Ready for High-Voltage Lithium Battery Applications? An Overview of Degradation Mechanisms and Battery Performance. *Adv Energy Mater* **12**, 2201264 (2022).
2. Meng N, Lian F, Cui G. Macromolecular Design of Lithium Conductive Polymer as Electrolyte for Solid-State Lithium Batteries. *Small* **17**, (2021).
3. Lopez J, Mackanic DG, Cui Y, Bao Z. Designing polymers for advanced battery chemistries. *Nat Rev Mater* **4**, 312-330 (2019).
4. Mackanic DG, *et al.* Decoupling of mechanical properties and ionic conductivity in supramolecular lithium ion conductors. *Nat Commun* **10**, 5384 (2019).
5. Wang Z, Chen C, Wang D, Zhu Y, Zhang B. Stabilizing Interfaces in High-Temperature NCM811-Li Batteries via Tuning Terminal Alkyl Chains of Ether Solvents. *Angew Chem Int Ed Engl*, e202303950 (2023).
6. Kohlmeyer RR, *et al.* Pushing the thermal limits of Li-ion batteries. *Nano Energy* **64**, 103927 (2019).
7. Zheng T, *et al.* Cocktail therapy towards high temperature/high voltage lithium metal battery via solvation sheath structure tuning. *Energy Storage Mater* **38**, 599-608 (2021).
8. Wang Z, Shen L, Deng S, Cui P, Yao X. 10 μm -Thick High-Strength Solid Polymer Electrolytes with Excellent Interface Compatibility for Flexible All-Solid-State Lithium-Metal Batteries. *Adv Mater* **33**, 2100353 (2021).
9. Wang H, *et al.* Thiol-Branched Solid Polymer Electrolyte Featuring High Strength, Toughness, and Lithium Ionic Conductivity for Lithium-Metal Batteries. *Adv Mater* **32**, e2001259 (2020).
10. Didwal PN, Verma R, Nguyen AG, Ramasamy HV, Lee GH, Park CJ. Improving Cyclability of All-Solid-State Batteries via Stabilized Electrolyte-Electrode Interface with Additive in Poly(propylene carbonate) Based Solid Electrolyte. *Adv Sci* **9**, 2105448 (2022).
11. Cui Y, Wan JY, Ye YS, Liu K, Chou LY. A Fireproof, Lightweight, Polymer-Polymer Solid-State Electrolyte for Safe Lithium Batteries. *Nano Lett* **20**, 1686-

1692 (2020).

12. Zheng Y, Li X, Li CY. A novel de-coupling solid polymer electrolyte via semi-interpenetrating network for lithium metal battery. *Energy Storage Mater* **29**, 42-51 (2020).
13. Wan J, *et al.* Ultrathin, flexible, solid polymer composite electrolyte enabled with aligned nanoporous host for lithium batteries. *Nat Nanotechnol* **14**, 705-711 (2019).

REVIEWERS' COMMENTS

Reviewer #1 (Remarks to the Author):

The authors satisfactorily addressed my and Reviewer #2's concerns so that I think this manuscript is ready for publication in Nature Communcations.